# FIGNL1 AAA+ ATPase remodels RAD51 and DMC1 filaments in pre-meiotic DNA replication and meiotic recombination

Masaru Ito [1] ✉, Asako Furukohri [1], Kenichiro Matsuzaki [1,2], Yurika Fujita[1], Atsushi Toyoda [3] & Akira Shinohara [1] ✉

The formation of RAD51/DMC1 filaments on single-stranded (ss)DNAs essential for homology search and strand exchange in DNA double-strand break (DSB) repair is tightly regulated. FIGNL1 AAA+++ ATPase controls RAD51-mediated recombination in human cells. However, its role in gametogenesis remains unsolved. Here, we characterized a germ line-specific conditional knockout (cKO) mouse of *FIGNL1*. *Fignl1* cKO male mice showed defective chromosome synapsis and impaired meiotic DSB repair with the accumulation of RAD51/DMC1 on meiotic chromosomes, supporting a positive role of FIGNL1 in homologous recombination at a post-assembly stage of RAD51/DMC1 filaments. *Fignl1* cKO spermatocytes also accumulate RAD51/DMC1 on chromosomes in pre-meiotic S-phase. These RAD51/DMC1 assemblies are independent of meiotic DSB formation. We also showed that purified FIGNL1 dismantles RAD51 filament on double-stranded (ds)DNA as well as ssDNA. These results suggest an additional role of FIGNL1 in limiting the non-productive assembly of RAD51/DMC1 on native dsDNAs during pre-meiotic S-phase and meiotic prophase I.

RAD51 is a central player in homologous recombination in eukaryotes by catalyzing homology-directed repair such as the repair of DNA double-strand breaks (DSBs) in both somatic and meiotic cells. RAD51 also plays a role in S-phase for the protection of stalled DNA replication forks[1,2], which is independent of RAD51-mediated strand exchange[3]. As an active form for homology search and strand exchange, multiple protomers of RAD51 bind to single-stranded (ss)DNAs in the presence of ATP to form a right-handed helical filament termed RAD51 nucleofilament, in which ssDNAs are extended 1.5–2.0-fold relative to the B-form DNA[4,5]. RAD51 nucleofilament is highly dynamic; its assembly and disassembly are controlled by multiple proteins/complexes. The RAD51 mediator promotes the assembly of RAD51 nucleofilament on the ssDNAs coated with an ssDNA-binding protein, RPA (Replication protein A), which hampers RAD51 binding to the ssDNA. The RAD51 mediator in humans includes BRCA2(-PALB2-DSS1)[6], BRCA1-BARD1[7],

RAD52[8], SWI5-SFR1/MEI5[9], and five RAD51 paralogs (RAD51B, -C, -D, XRCC2, and −3) in vertebrates including humans. The RAD51 paralogs form two distinct complexes: RAD51C-XRCC3 and RAD51B-RAD51C-RAD51D-XRCC2[10]. SWSAP1 is a distinct member of a RAD51 paralog[11,12], which forms a complex with SWS1 and SPIDR. At the post-assembly stage, RAD51 nucleofilament works together with the SWI2/SNF2 family, RAD54 and RAD54B[13], and RAD51AP1[14] for efficient homology search and strand exchange with a target double-stranded DNA (dsDNA) for the formation of displacement (D)-loop. RAD51-mediated strand invasion provides a template for DNA synthesis for further processing of the intermediates for the completion of the repair.

The dynamics of the RAD51 filament are also negatively regulated by proteins that disassemble the filament. Various DNA helicases are involved in the disassembly of RAD51 filament, which include BLM[15], FANCJ[16], RECQ5[17], and FBH1[18] as well as a degenerate DNA helicase,

[1]Institute for Protein Research, Osaka University, Suita, Osaka 565-0871, Japan. [2]Department of Advanced Bioscience, Graduate School of Agriculture, Kindai University, Nara, Nara 631-8505, Japan. [3]Advanced Genomics Center, National Institute of Genetics, Mishima, Shizuoka 411-8540, Japan. ✉ e-mail: msrito2@protein.osaka-u.ac.jp; ashino@protein.osaka-u.ac.jp

PARI[19]. RAD51-mediated invasion intermediates are also remodeled by other DNA helicases such as WRN, RTEL1, FANCM, and RECQ1[20]. RAD54 translocase also disassembles RAD51 from dsDNA in recombination intermediates[21]. The stabilization and destabilization of RAD51 filaments are associated with the choice of a homologous recombination pathway[22].

Meiosis is a specialized cell division that produces haploid gametes. Homologous recombination, specifically crossing-over, is essential for chromosome segregation of homologous chromosomes during meiosis I. To ensure the crossover (CO) formation in meiosis, highly programmed regulation is implanted on the recombination pathway[23]. A meiosis-specific RAD51 cousin, DMC1, is critical for homology search in meiotic recombination in many species[24]. The cooperation of RAD51 and DMC1 promotes homology search between homologous chromosomes rather than sister chromatids whose mechanism is one big enigma in the field[25]. RAD51/DMC1-mediated recombination requires meiosis-specific proteins as well as proteins playing a role in somatic cells. Knockout (KO) of the most of genes necessary for homologous recombination, *Rad51* and RAD51 mediators (*Brca2, Rad51B, -C, -D, Xrcc2,* and *−3*), in mice leads to embryonic lethality[26−28]. *Swsap1* (and *Sws1* and *Spidr*) KO mice are viable but are sterile, with the exception of *Spidr* KO female mice being sub-fertile[11,29]. Indeed, SWSAP1-SWS1-SPIDR is necessary for the efficient formation of RAD51/DMC1 focus, which is a cytologically detectable immuno-stained structure, in male meiosis[11,29,30]. Like RAD51 mediators, KO mice of some negative regulators such as BLM are also embryonically lethal[31] while *Rad54* KO mice are viable and fertile[32].

A recently identified recombination regulator, FIGNL1, encodes an AAA+ ATPase which belongs to microtubule severing proteins including Katanin, Spastin, and Fidgetin with an N-terminal microtubule-binding domain[33]. FIGNL1 was identified as a RAD51-interacting protein and is required for homologous recombination in human somatic cells[34]. FIGNL1 knockdown does not affect DSB-induced RAD51-focus formation[29,34], suggesting its role in the post-RAD51 assembly step. Despite its positive role in homologous recombination, FIGNL1 functions as a negative regulator for RAD51 assembly. FIGNL1 depletion rescues defective RAD51-focus formation in human cells depleted for SWSAP1 or SWS1, but not for RAD51C[29]. Moreover, purified FIGNL1 promotes the dissociation of RAD51 protein from ssDNAs, which is suppressed by SWSAP1[29]. In *Arabidopsis thaliana* meiosis, a mutant of *FIGL1*, FIGNL1 ortholog, as well as its partner, *FLIP/FIRRM*, shows increased frequencies of COs during meiosis, indicating a role as an anti-CO factor[35,36]. The role of FIGL1-FLIP as the anti-CO factor is also seen in rice meiosis[37]. Importantly, the *figl1* mutation can suppress defective RAD51/DMC1-focus formation in *brca2* mutant meiocytes, indicating antagonism of FIGL1 with the RAD51 mediator[38]. These suggest that RAD51 mediators might play a distinct function in different recombination pathways. Although FIGNL1 plays a role in homologous recombination by regulating the SWSAP1 complex in somatic cells[29], it remains unknown how FIGNL1 regulates recombination and works together with SWSAP1 during meiosis.

In this study, we analyzed the role of FIGNL1 in mice in meiosis, by constructing a germ-cell specific conditional knockout (cKO) mouse and found that *Fignl1* depletion in the testis led to defective spermatogenesis. Meiotic chromosome spreads from *Fignl1* cKO spermatocytes showed a two-fold increase in the numbers of RAD51 and DMC1 foci relative to controls, suggesting the role of FIGNL1 in the post-assembly stage of not only RAD51 filament but also DMC1 filament. This defective disassembly results in the impaired loading of proteins necessary for CO formation. We also found that FIGNL1 suppresses RAD51 and DMC1 assembly in pre-meiotic S-phase and early leptonema, which are independent of meiotic DSB formation. Moreover, *Fignl1* cKO ameliorates defective RAD51- and DMC1-focus formation in KO spermatocytes of *Swasp1*, a RAD51/DMC1 mediator, supporting the antagonistic role of FIGNL1 to SWSAP1 in the assembly of both RAD51 and DMC1. Together, our study establishes a dual role of FIGNL1 in mammalian male meiosis. The same conclusion was obtained by Baudat's group by characterizing the *Fignl1* cKO as well as *Firrm/Flip* cKO[39].

## Results

### *Fignl1* is essential for early mouse embryogenesis

To determine the functions of FIGNL1 in mice, we created a conditional allele of *Fignl1* gene, *Fignl1flox*, in which the exon 3 encoding an open reading frame of the full-length protein is flanked by the *loxP* sites (Fig. 1a) in C57BL/6 background. A null allele of the gene, *Fignl1Δ*, was obtained by breeding with *CAG-Cre+* mice that constitutively express Cre recombinase. *Fignl1Δ/+* heterozygous mice grew normally and did not show any apparent defects. Breeding between male and female of *Fignl1Δ/+* produced *Fignl1+/+* and *Fignl1Δ/+* pups, but no *Fignl1Δ/Δ* homozygous pups (0/186 from 38 litters), showing that *Fignl1Δ/Δ* mice are not viable. This is consistent with the reported lethality of *Fignl1* knockout (KO) mice in the International Mouse Phenotyping Consortium (IMPC) (https://www.mousephenotype.org). In a detailed analysis of different embryonic stages by the breeding of heterozygous parents, we could not obtain *Fignl1Δ/Δ* homozygous embryo at both E7.5 and E13.5 stages (0/29 and 0/37, respectively), showing very early lethality of mice homozygous for *Fignl1* deletion. Early embryonic lethality in mice homozygous for the homologous recombination genes has been reported; *Rad51*[28], *Brca2*[40], and Rad51 paralogs (*Rad51B, -C, -D, Xrcc2,* and *−3*)[27].

### *Fignl1* conditional knockout mice are defective in spermatogenesis

To know the role of FIGNL1 in male meiosis, we generated *Fignl1flox/Δ* mice harboring a germ line-specific Cre recombinase gene, *Stra8-Cre*. The *Stra8-Cre* gene ensured the expression of Cre not only in spermatocytes (meiotic cells) but also in spermatogonia[41]. *Fignl1flox/Δ Stra8-Cre+*, hereafter *Fignl1* conditional knockout (cKO), male mice were characterized for spermatogenesis. *Fignl1* cKO adult male mice showed normal body weight without any morphological defects but reduced sizes of the testis by ~50% relative to controls (*Fignl1+/+, Fignl1flox/Δ,* and *Fignl1+/+ Stra8-Cre+*), which is a characteristic of mutants with defective spermatogenesis (Fig. 1b, c). Western blotting analysis confirmed decreased amounts of FIGNL1 protein in whole testis extracts (Fig. 1d). The level of residual FIGNL1 protein is consistent with the efficiency of Stra8-Cre-mediated excision of *Fignl1flox* allele estimated by the transmission of *Fignl1Δ* allele from *Fignl1flox/+ Stra8-Cre+* male mice (91.3%; 116/127). Sperm counts were ~6-fold lower in *Fignl1* cKO relative to controls (Fig. 1e). Residual sperms in *Fignl1* cKO might come from *Fignl1flox/Δ* cells where Stra8-Cre recombinase failed excision of the *Fignl1flox* allele. Histological analysis showed that seminiferous tubules in *Fignl1* cKO testes contained decreased numbers of late spermatocytes and spermatids (Fig. 1f). These indicate a critical role of FIGNL1 in spermatogenesis. Consistent with this, FIGNL1 expression was detected in PLZF-positive spermatogonia and SYCP3-positive spermatocytes, but not in Sertoli cells and round spermatids (Supplementary Fig. 1a).

To analyze meiotic defects in *Fignl1* cKO male mice, we prepared surface-spread chromosomes of spermatocytes and immuno-stained for meiosis-specific chromosomal proteins, SYCP3(Scp3) and SYCP1(Scp1), which are components of lateral/axial elements and central region of the synaptonemal complex (SC), respectively[42,43]. In leptonema of control spermatocytes, SYCP3 shows thin-line staining structures that correspond to elongating chromosome axes (from short to long lines). In zygonema, regions of two thin chromosome axes are paired locally to form a thick line of SYCP3, where SYCP1 proteins are deposited for the assembly of the SC central region. From early zygonema to early pachynema, SC elongates along entire chromosomes, resulting in pachytene chromosome configuration with 19

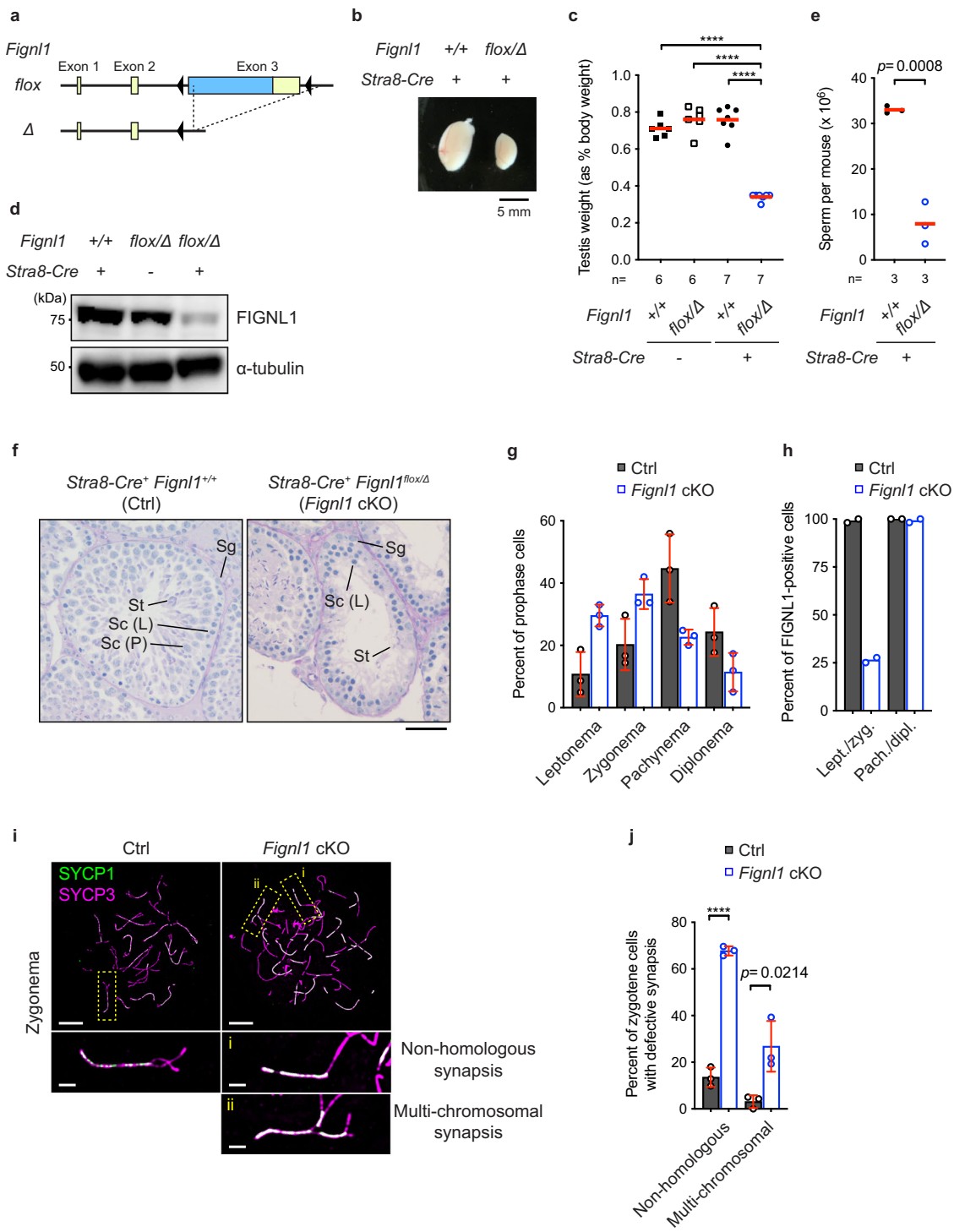

fully paired thick SYCP3 lines colocalized with SYCP1 and partially paired X-Y bivalent. In *Fignl1* cKO spermatocytes, like in controls, SYCP3-positive short and long lines were formed in leptonema, and short lines of SYCP1 in zygonema were also observed. However, there were fewer pachytene and diplotene cells in *Fignl1* cKO relative to controls ($22.6 \pm 2.4\%$ and $11.4 \pm 6.2\%$ compared to $44.7 \pm 11.0\%$ and $24.3 \pm 7.7\%$, respectively; Fig. 1g). We also evaluated the expression of FIGNL1 protein in testicular cell squashes by categorizing SYCP3-positive meiotic cells into two: cells before pachynema (i.e. leptonema and zygonema) with a pan-nuclear γH2AX staining and those after pachynema (i.e. pachynema and diplonema) with a dense γH2AX staining on the X-Y pair (the sex body)[44] (Supplementary Fig. 1b). In

controls, cells both before and after pachynema were nearly 100% positive for FIGNL1 staining. In contrast, while only ~25% of cells prior to pachynema were positive for FIGNL1, cells after pachynema were nearly 100% positive in *Fignl1* cKO (Fig. 1h). This indicates that *Fignl1*[Δ/Δ] cells are fully lost before pachynema from *Fignl1* cKO testes and only *Fignl1*[flox/Δ] cells where Stra8-Cre recombinase failed excision of the *Fignl1*[flox] allele progress further to complete spermatogenesis. Further analysis of surface-spread chromosomes revealed abnormal SC formation such as synapsis between non-homologous chromosomes ($67.7 \pm 2.0\%$ of cells compared to $13.6 \pm 4.1\%$ in controls) and multivalent synapsis ($26.8 \pm 10.9\%$ of cells compared to $3.1 \pm 2.7\%$ in controls) in *Fignl1* cKO (Fig. 1i, j), which could be defined as "pachytene-like" stage as seen in

**Fig. 1 | Defective spermatogenesis in *Fignl1* cKO mice. a** Scheme of mouse *Fignl1* *flox* and *Δ* alleles. Yellow and blue boxes and black triangles represent untranslated regions (UTRs), coding sequence (CDS), and *loxP* sequences, respectively. **b** Representative image of testes from mice with indicated genotypes. **c** Testis weights of adult mice (9–20 weeks) with indicated genotypes. The red bars are means. **d** Immunoblotting of FIGNL1 from whole-testis extracts of mice with indicated genotypes. α-tubulin is a loading control. **e** Sperm numbers of mice with indicated genotypes. The red bars are means. **f** Representative images of periodic acid-Schiff (PAS)-stained seminiferous tubule sections from the testis of *Fignl1*[+/+] *Stra8-Cre*[+] (Ctrl) and *Fignl1*[flox/Δ] *Stra8-Cre*[+] (*Fignl1* cKO) mice. Sg, spermatogonia; Sc (L), leptotene spermatocytes; Sc (P), pachytene spermatocytes; St, spermatids. **g** Distribution of different meiotic prophase I stages in Ctrl (gray bars) and *Fignl1* cKO (blue open bars). >100 cells were counted from each animal. The bar graphs indicate means. The red bars are SDs from three animals of each genotype. **h** Distribution of FIGNL1-positive and -negative spermatocytes in Ctrl (gray bars)

and *Fignl1* cKO (blue open bars). >100 leptotene/zygotene (lept./zyg.) and pachytene/diplotene (pach./dipl.) cells were analyzed from each animal. The means of two mice of each genotype are indicated. Representative images of testicular cell squashes are shown in Supplementary Fig. 1b. **i** Representative images of spermatocyte-chromosome spreads immunostained for SYCP1 (green) and SYCP3 (magenta) at zygonema in Ctrl and *Fignl1* cKO. The bottom panels are magnified images of regions with yellow dotted rectangles. **j** Quantification of the frequency of synapsis defects in Ctrl (gray bars) and *Fignl1* cKO (blue open bars). >90 cells were analyzed from each animal. The bar graphs indicate means. The red bars are SDs from three mice of each genotype. In (**f**)–(**i**), genotypes of indicated animals are Ctrl, *Fignl1*[+/+] *Stra8-Cre*[+]; *Fignl1* cKO, *Fignl1*[flox/Δ] *Stra8-Cre*[+]. The results of two-tailed unpaired *t*-tests are indicated in the graphs: ****$p \leq 0.0001$. Scale bars, 50 μm in (**f**) and 10 μm for whole-nucleus images and 2 μm for magnified images in (**i**). Source data are provided as a Source Data file.

other meiotic recombination defective mutants such as *Dmc1* and *Meiob* KO, etc.[45,46]. These results show that FIGNL1 is necessary for full synapsis of homologous chromosomes in late zygonema and/or early pachynema and that most of the FIGNL1-depleted cells seem to die at these stages or do not progress further.

### Compromised meiotic recombination in *Fignl1* cKO

ssDNA-binding proteins such as RPA and MEIOB-SPATA22 play a role in the early/middle stage of meiotic recombination both before and after the RAD51 and DMC1 assembly. To know the effect of *Fignl1* cKO on meiotic recombination, we examined the localization of RPA2, a middle-subunit of RPA, on chromosome spreads of spermatocytes, together with SYCP3 as a marker for the progression of meiotic prophase I. In control spermatocytes, RPA2 foci appear from leptonema with 200-300 foci on average in mid/late leptonema ($223 \pm 73$ foci; Fig. 2a, b). The number of RPA2 foci increased up to ~400 foci in zygonema ($376 \pm 45$ foci in midzygonema) and persisted throughout zygonema ($332 \pm 59$ foci in late zygonema). The RPA2-focus number in *Fignl1* cKO mid/late leptonema was increased up to 300-400 foci ($368 \pm 100$ foci) and a similar number persisted in the early zygotene stage ($409 \pm 59$ foci). The number of RPA2 foci in *Fignl1* cKO gradually decreased from midzygonema and was significantly lower than controls in late zygonema ($281 \pm 48$ foci compared to $332 \pm 59$ in controls; $p < 0.0001$, Mann–Whitney *U*-test; Fig. 2b), suggesting a role of FIGNL1 in the dynamic behavior of an RPA-associated recombination event. We also analyzed the histone γH2AX staining as a marker of DSB formation and its repair. Interestingly, *Fignl1* cKO spermatocytes with increased RPA2 foci showed a similar or slightly reduced intensity of γH2AX from leptonema to the late zygonema stage compared to controls (Supplementary Fig. 2a, b).

In meiotic recombination, previous cytological analyses showed that RPA2 foci persist longer than DMC1/RAD51 foci from mid/late zygonema to early/mid pachynema, suggesting a role of RPA in a late stage of meiotic recombination[47]. In zygonema (from early zygonema), foci containing the MutSγ complex with MSH4 and MSH5, which regulate recombination events for CO formation, appear specifically on synapsed regions and show similar behavior to RPA on meiotic chromosomes[48]. We analyzed MSH4-focus formation and found that the number of MSH4 foci in *Fignl1* cKO was lower than in controls in mid/late zygonema ($50 \pm 22$ foci versus $107 \pm 47$ foci in controls; Fig. 2c, d). Given the possibility that the reduction in MSH4-focus number is an indirect consequence of synapsis defects in *Fignl1* cKO, we also analyzed the density of MSH4 foci per unit length of SC. MSH4-focus number and SC length (total SYCP1 length) showed a linear relationship in control mid-zygonema ($r = 0.89$; Fig. 2e). In *Fignl1* cKO, a correlation between MSH4-focus number and SC length was also positive, but lower than in controls ($r = 0.58$). Importantly, the density of MSH4 foci per unit length of SC is reduced in *Fignl1* cKO relative to controls ($0.73 \pm 0.22$ foci per μm of SC versus $0.98 \pm 0.18$ foci per μm

of SC in controls; Fig. 2f). These suggest that FIGNL1 is required for efficient loading of MSH4 for meiotic recombination.

CO formation is a key outcome of meiotic recombination. The CO sites on meiotic chromosomes are marked with the foci of MutLγ complex such as MLH1[49]. Control late pachytene cells show ~25 MLH1 foci per nucleus. On the other hand, we could not detect any MLH1 foci on late zygotene/pachytene-like chromosomes of *Fignl1* cKO spermatocytes (Supplementary Fig. 2c). This is consistent with the observation that FIGNL1 is essential for the progression of spermatocytes into a pachytene stage with MLH1 foci.

### FIGNL1 promotes the disassembly of DMC1 in male meiosis

We next examined the localization of DMC1 on meiotic chromosomes. As shown previously[50–52], in control spermatocytes, DMC1 foci appear in leptonema, peak in their number at early zygonema, and gradually decrease throughout zygonema (Fig. 3a, b). In *Fignl1* cKO, DMC1 foci followed similar kinetics as seen in controls. However, at any stages, the number of DMC1 foci in *Fignl1* cKO was more than that in controls. In early zygonema, ~500 DMC1 foci ($449 \pm 89$ foci) are observed in *Fignl1* cKO, which is 2.3-fold more than in controls ($194 \pm 32$ foci). DMC1-focus number in *Fignl1* cKO clearly increased through meiotic prophase I progression. In late zygonema, the number of DMC1 foci in *Fignl1* cKO was reduced by half of that in early zygonema ($220 \pm 70$ foci compared to $449 \pm 89$ foci), indicating a turnover of the foci. These results suggest that FIGNL1 suppresses excess DMC1-focus formation and/or timely progression of DMC1-mediated recombination.

In *Fignl1* cKO spermatocytes, we always observed two populations: cells with a high number of DMC1 foci and cells with a similar number of foci to controls. We assumed that the latter is *Fignl1*[flox/Δ] cells where Cre recombinase failed excision of the *Fignl1*[flox] allele into *Fignl1*[Δ]. Note that we included these foci for our quantification and that this population becomes an internal control for our analysis.

### FIGNL1 promotes the disassembly of RAD51 in spermatocytes

In human cultured cells, FIGNL1 regulates RAD51 assembly[29,34]. We examined the localization of RAD51 on meiotic chromosomes. As reported previously[50–52], in control spermatocytes, RAD51 foci are observed from mid/late leptonema and peak in early zygonema, and then decrease toward pachynema (Fig. 3c, d). The average number of RAD51 foci is $107 \pm 56$ and $181 \pm 27$ foci per nucleus in control mid/late leptotene and early zygotene stages, respectively. As seen for DMC1, we detected an increased number of RAD51 foci in *Fignl1* cKO cells. The average number of RAD51 foci in early zygonema is $380 \pm 98$ foci, which is more than twice compared to that of controls ($181 \pm 27$ foci). The number of RAD51 foci in zygonema is similar to that of DMC1 foci, with ~600 RAD51 foci in some early zygotene nuclei. When the zygotene stage progressed from early to late, the RAD51-focus number in *Fignl1* cKO gradually decreased as in controls. However, even at late zygonema, we detected a significant increase in RAD51-focus number

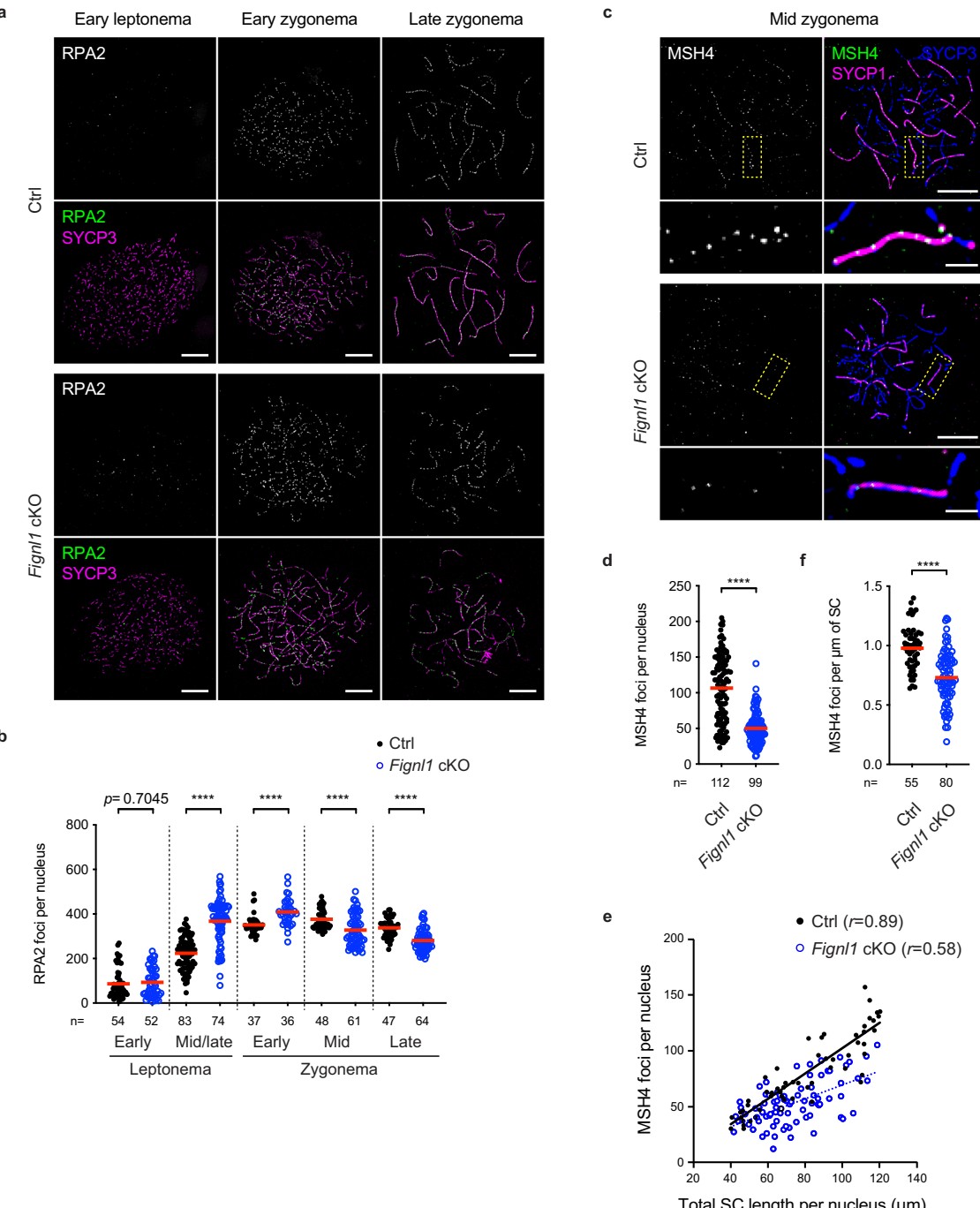

**Fig. 2 | Deregulated meiotic recombination in *Fignl1* cKO spermatocytes.**
**a** Representative images of spermatocyte-chromosome spreads immunostained for RPA2 (white in the top panels and green in the bottom panels) and SYCP3 (magenta) at indicated meiotic prophase I stages in Ctrl and *Fignl1* cKO. **b** Quantification of RPA2 focus numbers at different meiotic prophase I stages in Ctrl (black circles) and *Fignl1* cKO (blue open circles). The red bars are means. **c** Representative images of spermatocyte-chromosome spreads immunostained for MSH4 (white in the left panels and green in the right panels), SYCP1 (magenta) and SYCP3 (blue) at mid-zygonema in Ctrl and *Fignl1* cKO. The bottom panels are magnified images of regions with dotted yellow. **d** Quantification of MSH4 focus numbers at mid-zygonema in Ctrl (black circles) and *Fignl1* cKO (blue open circles). The red bars are means. **e** The

correlation between MSH4 focus number and total SC length at midzygonema in Ctrl (black circles) and *Fignl1* cKO (blue open circles). Pearson's *r* is shown in parentheses. A black solid line and a blue dashed line indicate linear regression for Ctrl and *Fignl1* cKO, respectively. **f** Quantification of MSH4 focus density on the SCs at mid-zygonema in Ctrl (black circles) and *Fignl1* cKO (blue open circles). The red bars are means. In (**e**) and (**f**), midzygotene nuclei with 40–120 μm of total SC length per nucleus were analyzed. Genotypes of indicated animals are: Ctrl, *Fignl1^{+/+} Stra8-Cre^+*; *Fignl1* cKO, *Fignl1^{flox/Δ} Stra8-Cre^+*. The results of the two-tailed Mann–Whitney *U*-test are indicated in the graphs: ****$p \leq 0.0001$. The total number of cells analyzed is indicated below the graphs. Scale bars in (**a**) and (**c**), 10 μm and 2 μm for magnified images in (**c**). Source data are provided as a Source Data file.

in *Fignl1* cKO relative to controls ($182 \pm 86$ foci compared to $128 \pm 32$ foci in controls; $p = 0.0163$, Mann–Whitney *U*-test; Fig. 2d). These results show that FIGNL1 promotes the timely resolution of RAD51 ensembles during the zygotene stage.

We note that the focus intensities of RAD51 and DMC1 in *Fignl1* cKO spermatocytes were slightly decreased compared to controls (Supplementary Fig. 3), suggesting that in *Fignl1* cKO spermatocytes, more sites for the accumulation of RAD51/DMC1 molecules are

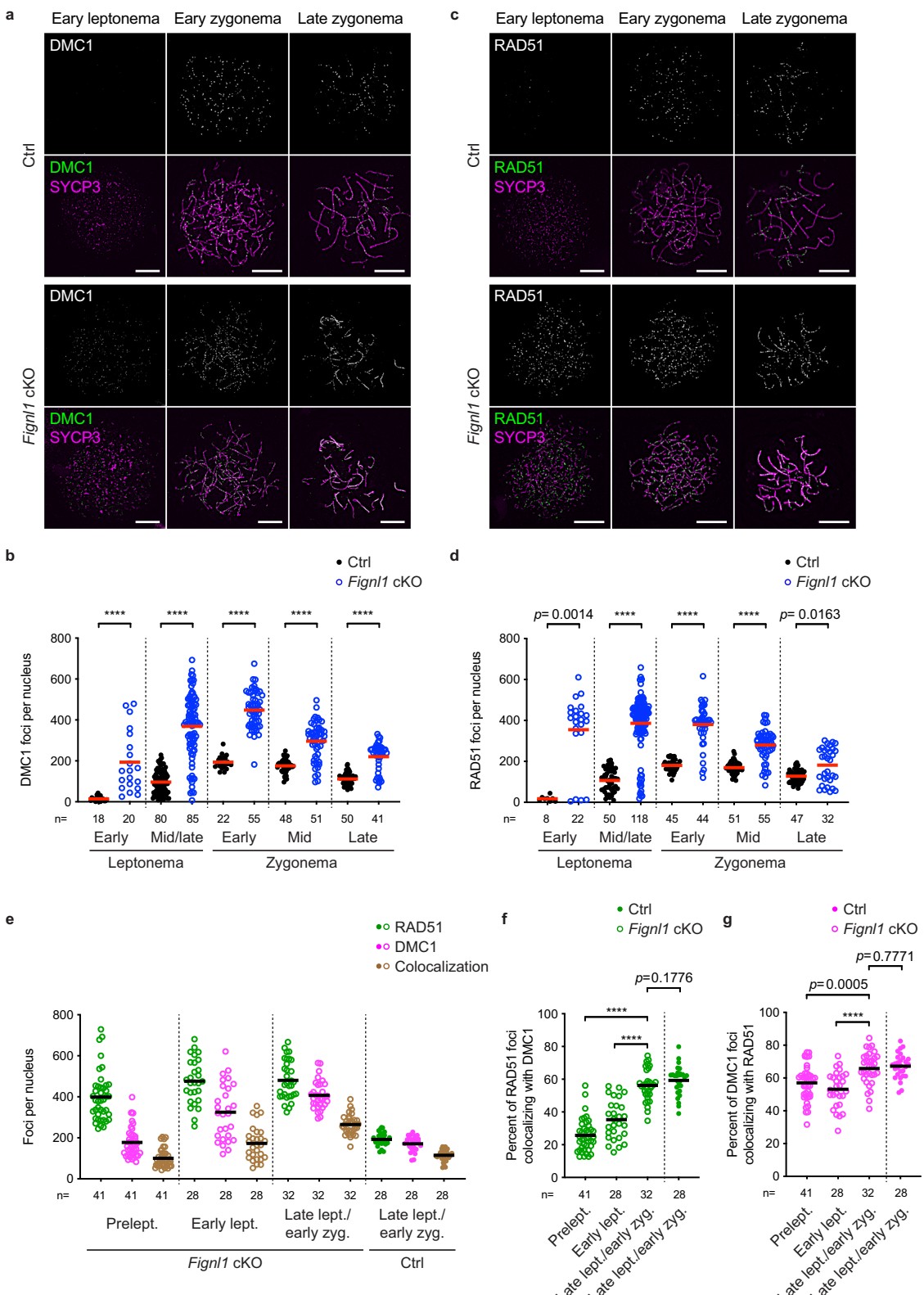

present, fewer molecules of RAD51/DMC1 protein are on each accumulation site because of a limited number of RAD51/DMC1 protein in a cell.

**Increased RAD51/DMC1 foci in early leptonema in *Fignl1* cKO**

In contrast to the zygotene stage where the numbers of DMC1 and RAD51 foci are comparable in both controls and *Fignl1* cKO, focus

numbers of DMC1 and RAD51 were clearly different at early leptonema only in *Fignl1* cKO (Fig. 3a–d). In mid/late leptonema when DMC1 and RAD51 foci appear on elongating SYCP3 stretches in controls, the number of DMC1 foci in *Fignl1* cKO is 369 ± 156 foci per cell, which is about 3–4 times more of that in controls (96 ± 57 foci). The number of RAD51 foci in mid/late leptonema in *Fignl1* cKO is 387 ± 134 foci, 3-4 times more than in controls (107 ± 56 foci). In early leptonema, where

**Fig. 3 | Accumulation of RAD51 and DMC1 in meiotic prophase I in *Fignl1* cKO spermatocytes. a** Representative images of spermatocyte-chromosome spreads immunostained for DMC1 (white in the top panels and green in the bottom panels) and SYCP3 (magenta) at indicated meiotic prophase I stages in Ctrl and *Fignl1* cKO. **b** Quantification of DMC1 focus numbers at different meiotic prophase I stages in Ctrl (black circles) and *Fignl1* cKO (blue open circles). The red bars are means. **c** Representative images of spermatocyte-chromosome spreads immunostained for RAD51 (white in the top panels and green in the bottom panels) and SYCP3 (magenta) at indicated meiotic prophase I stages in Ctrl and *Fignl1* cKO. **d** Quantification of RAD51 focus numbers at different meiotic prophase I stages in Ctrl (black circles) and *Fignl1* cKO (blue open circles). The red bars are means. **e** Quantification of focus numbers of RAD51 (green), DMC1 (magenta), and RAD51-

DMC1 colocalization (brown) at different meiotic prophase I stages in Ctrl (filled circles) and *Fignl1* cKO (open circles). The black bars are means. Representative images of spermatocyte-chromosome spreads are shown in Supplementary Fig. 7. **f, g** Quantification of the frequency of RAD51-DMC1 colocalization at different meiotic prophase I stages in Ctrl (filled circles) and *Fignl1* cKO (open circles). The degrees of RAD51 foci colocalizing with DMC1 (**f**, green) and DMC1 foci colocalizing with RAD51 (**g**, magenta) are shown. The black bars are means. Genotypes of indicated animals are: Ctrl, *Fignl1*<sup>+/+</sup> *Stra8-Cre*<sup>+</sup>; *Fignl1* cKO, *Fignl1*<sup>flox/Δ</sup> *Stra8-Cre*<sup>+</sup>. The results of the two-tailed Mann−Whitney *U*-test are indicated in the graphs: ****$p \le 0.0001$. The total number of cells analyzed is indicated below the graphs. Scale bars in (**a**) and (**c**), 10 μm. Prelept., preleptonema; lept., leptonema; zyg., zygonema. Source data are provided as a Source Data file.

SYCP3 short stretches are yet to be elongated, DMC1 and RAD51 foci rarely appear in control spermatocytes. However, *Fignl1* cKO formed a relatively larger number of RAD51 foci than DMC1: the number of RAD51 foci was $355 \pm 178$, which exceeds the number of DMC1 foci ($194 \pm 148$ foci) in early leptonema. These suggest that FIGNL1 suppresses the formation of excess numbers of RAD51 and DMC1 foci in early meiosis I, and that, in the absence of FIGNL1, RAD51 and DMC1 loading is uncoupled (see below).

### Pre-meiotic S-phase in *Fignl1* cKO accumulates RAD51/DMC1 foci

In wild-type meiosis, the focus formation of both RAD51 and DMC1 is mediated by SPO11-induced DSBs in leptonema[53,54]. The increased number of RAD51 and DMC1 foci on the early leptotene chromosome of *Fignl1* cKO spermatocytes led us to analyze preleptotene (pre-meiotic S-phase) spermatocytes in order to characterize the origin of excess RAD51/DMC1 foci that form in the absence of FIGNL1. For in vitro labeling of DNA replication, we incubated testis cell suspension with EdU for 1 h in a culture medium. The preleptotene spermatocytes, where a limited number of SYCP3 short stretches forms along with a numerous number of "tiny" SYCP3 dots, were categorized into two classes: pan dotty staining of EdU in early preleptonema and clustered dotty staining of EdU in mid/late preleptonema. In controls, both early and mid/late preleptotene EdU-positive spermatocytes showed very few foci of RAD51 (Fig. 4a, b). On the other hand, we could detect RAD51 foci in EdU-positive spermatocytes of *Fignl1* cKO, indicating RAD51-focus formation during pre-meiotic S phase. The number of RAD51 foci during pre-meiotic S-phase is around 300 in focus-positive nuclei ($298 \pm 110$ foci and $337 \pm 108$ foci in early and mid/late pre-leptonema, respectively), which is ~3/4 of the RAD51-focus number in meiotic prophase I of *Fignl1* cKO mice, but ~1.5-fold more of that in early zygonema of control mice (compare Figs. 3d and 4b). Some RAD51 foci in pre-meiotic S-phase are colocalized with EdU dots/lines ($38.8 \pm 12.6\%$ of RAD51 foci colocalized with EdU; Supplementary Fig. 4e), suggesting that RAD51 foci are associated with DNA replication. However, when DNA replication labeling with another thymidine analog IdU was shortened for 10 min, most of RAD51 foci did not colocalize with IdU foci (Supplementary Fig. 4a–d), though the degree of colocalization was significantly higher than random colocalization ($3.7 \pm 1.9\%$ of IdU foci colocalized with RAD51 compared to $2.5 \pm 1.3\%$, $p = 0.0261$, Mann−Whitney *U*-test; $9.1 \pm 4.5\%$ of RAD51 foci colocalized with IdU compared to $6.2 \pm 3.3\%$, $p = 0.0439$, Mann−Whitney *U*-test; Supplementary Fig. 4c, d). It is unlikely that RAD51 is associated with ongoing DNA replication forks in *Fignl1* cKO spermatocytes.

Further evaluation of DMC1 localization in *Fignl1* cKO revealed the presence of preleptotene spermatocytes containing DMC1 foci (Fig. 4c, d). The number of DMC1 foci in focus-positive nuclei is around 150 ($136 \pm 71$ foci and $164 \pm 70$ foci in early and mid/late preleptonema, respectively), about half of RAD51 foci in *Fignl1* cKO. As seen for RAD51, most of DMC1 foci did not colocalize with ongoing DNA replication forks when DNA replication was labeled with EdU for 10 min ($23.0 \pm 7.9\%$ of DMC1 foci colocalized with EdU;

Supplementary Fig. 5a–d). Of note is that very few DMC1 foci were detected in control preleptotene spermatocytes. This suggests that FIGNL1 limits the assembly of mitotic recombination machinery with RAD51 on chromatin during pre-meiotic S-phase as well as meiotic recombination machinery containing DMC1.

In contrast to compromised meiotic recombination and homolog synapsis as described above, pre-meiotic DNA replication in *Fignl1* cKO seemed largely normal, as the number of RPA2 foci, which are expected to correspond to replication fork-associated ssDNAs, and IdU foci with 10 min labeling, which indicates ongoing DNA replication forks, in early preleptotene spermatocytes were comparable between *Fignl1* cKO and controls ($624 \pm 241$ RPA2 foci in *Fignl1* cKO compared to $607 \pm 217$ foci in controls; $p = 0.9414$, Mann-Whitney *U*-test; $500 \pm 131$ IdU foci in *Fignl1* cKO compared to $476 \pm 140$ foci in controls; $p = 0.4652$, Mann-Whitney *U*-test; Supplementary Fig. 6a–c). However, we cannot exclude the possibility that FIGNL1 plays a role in pre-meiotic DNA replication as recently shown in *FIGNL1* KO human cell lines[55].

### Uncoupled loading of RAD51 and DMC1 in *Fignl1* cKO spermatocytes

Despite the overall increase in RAD51- and DMC1-focus numbers at each analyzed stage of *Fignl1* cKO spermatocytes, we observed ~2-fold more RAD51 foci compared to DMC1 foci in preleptotene and early leptotene stages, suggesting uncoupling of RAD51 and DMC1 loading onto chromosomes in the absence of FIGNL1. Co-staining of RAD51 and DMC1 confirmed different kinetics of RAD51 and DMC1 in *Fignl1* cKO (Fig. 3e, Supplementary Fig. 7); DMC1-focus numbers gradually increase from preleptonema to late leptonema/early zygonema ($177 \pm 74$, $325 \pm 136$, and $407 \pm 69$ foci in preleptonema, early lepto-nema, and late leptonema/early zygonema, respectively; Fig. 3e), whereas RAD51-focus numbers are nearly constant in those stages ($399 \pm 120$, $476 \pm 109$, and $480 \pm 95$ foci in preleptonema, early lepto-nema, and late leptonema/early zygonema, respectively). Colocalization analysis revealed a gradual increase in the number of RAD51-DMC1 co-foci as DMC1 foci increased. In preleptotene and early leptotene spermatocytes, 50-60% of DMC1 foci colocalized with RAD51 foci ($57.0 \pm 10.6\%$ and $53.1 \pm 11.0\%$ in preleptonema and early leptonema, respectively; Fig. 3g) and <40% of RAD51 foci colocalized with DMC1 foci ($25.7 \pm 9.9\%$ and $35.3 \pm 12.0\%$ in preleptonema and early lepto-nema, respectively; Fig. 3f). The degree of colocalization at these stages is significantly lower than that in the late leptotene/early zygotene stage where comparable levels of colocalization with controls were observed ($65.8 \pm 10.1\%$ and $67.2 \pm 7.6\%$ of DMC1 foci colocalized with RAD51 in *Fignl1* cKO and controls, respectively; $56.2 \pm 9.6\%$ and $59.3 \pm 8.9\%$ of RAD51 foci colocalized with DMC1 foci in *Fignl1* cKO and controls, respectively; Fig. 3f, g). These suggest that coupled loading of RAD51 and DMC1 in pre-meiotic S-phase and early meiotic prophase I is compromised in *Fignl1* cKO spermatocytes. On the other hand, the fact that more than half of DMC1 foci are colocalized with RAD51 in preleptonema suggests that DMC1 preferentially binds to meiotic chromatin where RAD51 is already bound.

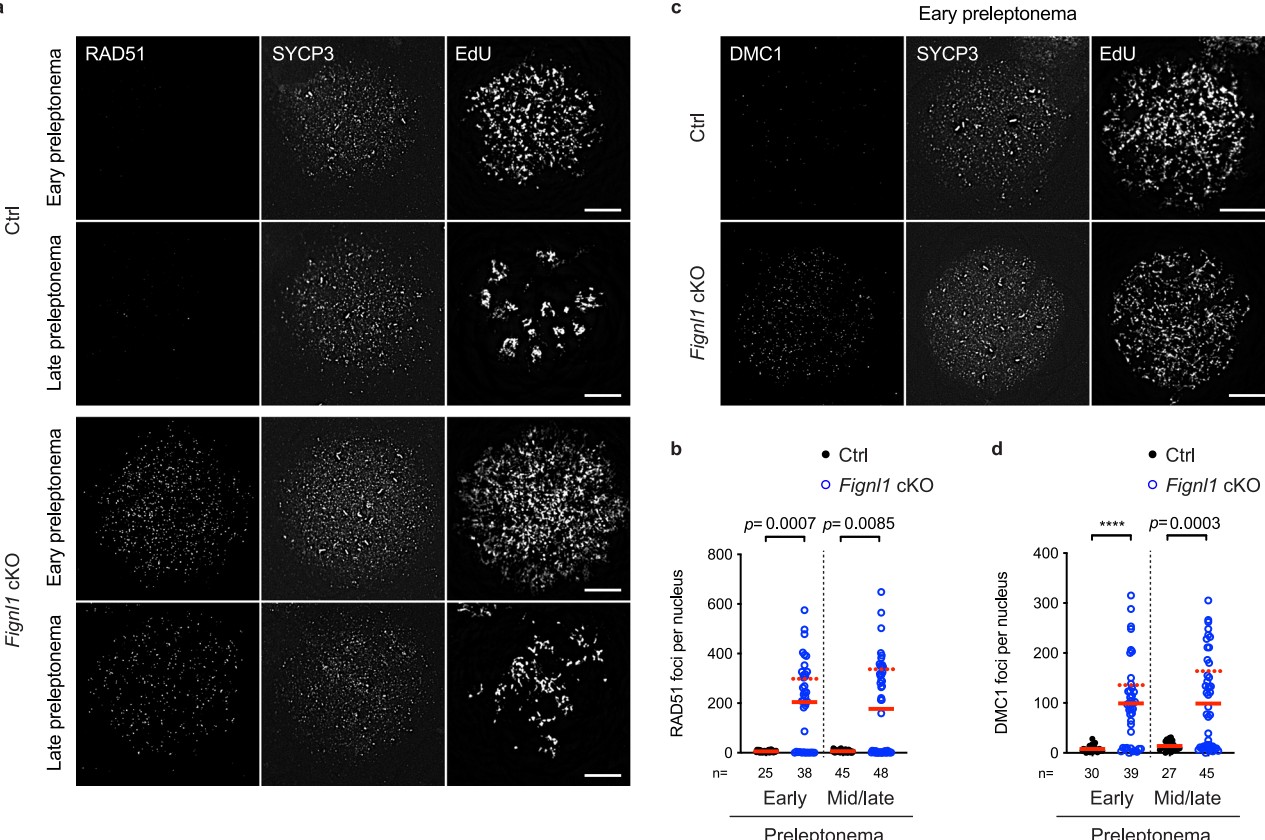

**Fig. 4 | Accumulation of RAD51 and DMC1 in preleptotene *Fignl1* cKO spermatocytes. a** Representative images of spermatocyte-chromosome spreads immunostained for RAD51 (left), SYCP3 (middle) and EdU (right, 60′ labeling) at preleptotene in Ctrl and *Fignl1* cKO. **b** Quantification of RAD51 focus numbers at preleptotene in Ctrl (black circles) and *Fignl1* cKO (blue open circles). The red solid bars and dashed bars are means for total cells and RAD51-positive cells (with >10 RAD51 foci; 26/38 and 25/48 cells in early and mid/late preleptotene, respectively), respectively. **c** Representative images of spermatocyte-chromosome spreads immunostained for DMC1 (left), SYCP3 (middle), and EdU (right, 60′

labeling) at preleptotene in Ctrl and *Fignl1* cKO. **d** Quantification of DMC1 focus numbers at preleptotene in Ctrl (black circles) and *Fignl1* cKO (blue open circles). The red solid bars and dashed bars are means for total cells and DMC1-positive cells (with >30 DMC1 foci; 28/39 and 26/45 cells in early and mid/late preleptotene, respectively), respectively. Genotypes of indicated animals are: Ctrl, *Fignl1*[+/+] *Stra8-Cre*[+]; *Fignl1* cKO, *Fignl1*[flox/Δ] *Stra8-Cre*[+]. The results of the two-tailed Mann-Whitney *U*-test are indicated in the graphs: ****$p \leq 0.0001$. The total number of cells analyzed is indicated below the graphs. Scale bars in (**a**) and (**c**), 10 µm. Source data are provided as a Source Data file.

## RAD51/DMC1-focus formation in *Spo11* KO *Fignl1* cKO spermatocytes

Both RAD51 and DMC1 foci appear in pre-meiotic S-phase in *Fignl1* cKO spermatocytes, which is much earlier than meiotic prophase I with programmed DSB formation by SPO11. To examine whether RAD51/DMC1-focus formation in *Fignl1* cKO mice is independent of meiotic DSB formation or not, we constructed *Spo11* KO *Fignl1* cKO mice (*Spo11*[−/−] *Fignl1*[flox/Δ] *Stra8-Cre*[+]), which are expected to be defective in SPO11-mediated DSB formation[53,54]. As shown previously, spermatocytes from control *Spo11* KO mice (*Spo11*[−/−] *Fignl1*[flox/Δ]) did not show any RAD51 foci on meiotic chromosomes (Fig. 5a, b, Supplementary Fig. 8a). On the other hand, spermatocytes from *Spo11* KO *Fignl1* cKO show on average 300-400 RAD51 foci from leptotene to zygotene(-like) stages ($407 \pm 77$, $390 \pm 64$, and $292 \pm 34$ foci for focus-positive nuclei in leptonema, early zygonema, and late zygonema, respectively; Fig. 5b). Moreover, *Spo11* KO *Fignl1* cKO also accumulates DMC1 foci with similar numbers to RAD51 foci ($374 \pm 108$, $391 \pm 57$, and $342 \pm 54$ foci for focus-positive nuclei in leptonema, early zygonema, and midzygonema, respectively; Fig. 5c, d, Supplementary Fig. 8b). Comparable numbers of RAD51 and DMC1 foci between *Fignl1* cKO and *Spo11* KO *Fignl1* cKO indicate that a large fraction of RAD51 and DMC1 foci on meiotic chromatin in the absence of FIGNL1 is independent of SPO11, thus meiotic DSBs. Like *Spo11* KO, *Spo11* KO *Fignl1* cKO spermatocytes lack pan-nuclear γH2AX staining and form few RPA2 foci

(Supplementary Fig. 8c–f), supporting the assembly of RAD51/DMC1 complexes in the absence of endogenous DSBs and ssDNAs. *Spo11* KO *Fignl1* cKO spermatocytes died in the pachytene-like stage with the formation of the pseudo-sex body (Supplementary Fig. 8c) as seen in *Spo11* KO spermatocytes[53,54]. We also detected RAD51/DMC1 foci in preleptotene spermatocytes of *Spo11* KO *Fignl1* cKO mice (Fig. 5f). This confirms that the formation of RAD51 and DMC1 foci during pre-meiotic S-phase is independent of the SPO11 function. Importantly, the degree of RAD51-DMC1 colocalization from preleptonema to early zygonema in *Spo11* KO *Fignl1* cKO is comparable to that in *Fignl1* cKO and more than half of DMC1 foci are colocalized with RAD51 at those stages in *Spo11* KO *Fignl1* cKO (Fig. 5e–h). This supports the idea that DMC1 preferentially binds to RAD51-bound meiotic chromatin, which is most likely dsDNA in the absence of meiotic DSB formation.

## RAD51 binds to and persists on meiotic recombination sites in *Fignl1* cKO spermatocytes

As described above, the absence of FIGNL1 increases the SPO11-independent formation of RAD51 and DMC1 foci in meiotic prophase I, some of which would come from pre-meiotic S-phase. Comparable numbers of RAD51 and DMC1 foci in the presence and absence of SPO11 (i.e., meiotic DSBs) prompted us to check whether RAD51/DMC1 can bind to meiotic recombination hotspots, especially ssDNA. Chromatin immunoprecipitation followed by single-stranded DNA

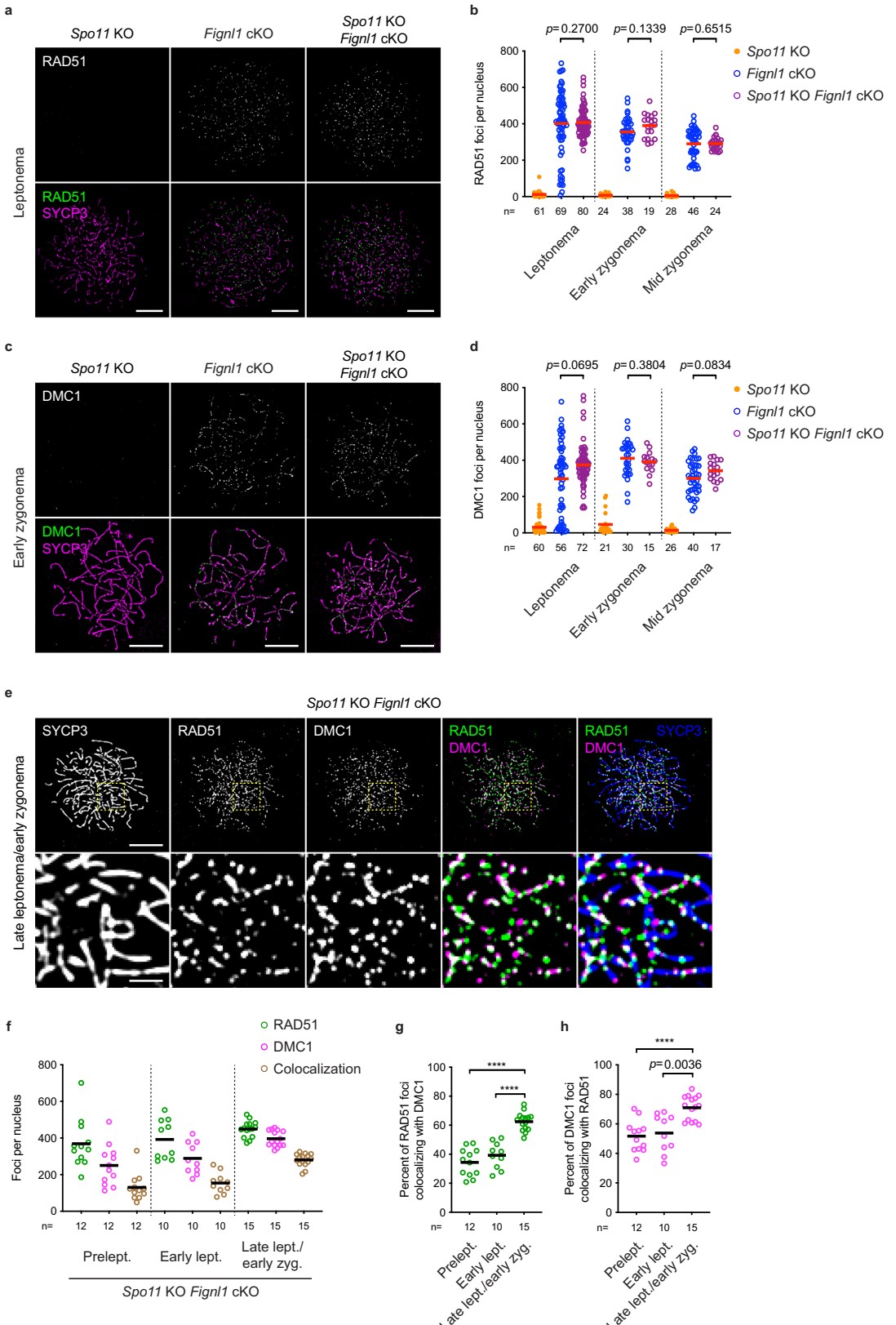

sequencing (ChIP-SSDS) for RAD51[56–58] revealed that in *Fignl1* cKO testes, like in controls, RAD51 binds to recombination hotspots where DMC1 strongly binds in wild-type testes (Fig. 6a, Supplementary Fig. 9a). Importantly, when RAD51 binding around hotspots is ranked based on the number of SPO11-oligos in wild-type, a similar binding profile of RAD51 is seen between *Fignl1* cKO and controls, with a small

increase in RAD51 ChIP-SSDS signals in two out of three *Fignl1* cKO mice (Fig. 6b, Supplementary Fig. 9b–d). We also ran RAD51 ChIP-SSDS in *Spo11* KO and *Spo11* KO *Fignl1* cKO testes, confirming no RAD51 enrichment at recombination hotspots and lower RAD51 enrichment in the entire genome (Supplementary Fig. 10a–d). Most of non-overlapping 2-kb bins with high RAD51 enrichment in *Spo11* KO *Fignl1*

**Fig. 5 | SPO11-independent accumulation of RAD51 and DMC1 in *Fignl1* cKO spermatocytes. a**, **c** Representative images of spermatocyte-chromosome spreads immunostained for RAD51 (**a**), DMC1 (**c**) (white in the top panels and green in the bottom panels) and SYCP3 (magenta) at leptonema in *Spo11* KO, *Fignl1* cKO and *Spo11* KO *Fignl1* cKO. **b**, **d** Quantification of RAD51 (**b**) and DMC1 (**d**) focus numbers at different meiotic prophase I stages in *Spo11* KO (orange circles), *Fignl1* cKO (blue open circles), and *Spo11* KO *Fignl1* cKO (purple open circles). The red bars are means. Cells in which Stra8-Cre recombinase seemed to fail excision of the *Fignl1*$^\Delta$ allele in *Spo11* KO *Fignl1* cKO (<30 RAD51 foci or <60 DMC1 foci) are excluded. Quantification including these cells is shown in Supplementary Fig. 8a, b. **e** Representative images of spermatocyte-chromosome spreads immunostained for RAD51 (green), DMC1 (magenta), and SYCP3 (blue) at late zygonema/early leptonema in *Spo11* KO *Fignl1* cKO. The bottom panels are magnified images of regions with yellow dotted rectangles. **f** Quantification of focus numbers of RAD51 (green open circles), DMC1 (magenta open circles), and RAD51-DMC1 colocalization (brawn open circles) at different meiotic prophase I stages in *Spo11* KO *Fignl1* cKO. The black bars are means. **g**, **h** Quantification of the frequency of RAD51-DMC1 colocalization at different meiotic prophase I stages in *Spo11* KO *Fignl1* cKO. The degrees of RAD51 foci colocalizing with DMC1 (**g**, green open circles) and DMC1 foci colocalizing with RAD51 (**h**, magenta open circles) are shown. The black bars are means. Genotypes of indicated animals are: *Spo11* KO, *Spo11*$^{-/-}$ *Fignl1*$^{flox/\Delta}$; *Fignl1* cKO, *Fignl1*$^{flox/\Delta}$ *Stra8-Cre*$^+$; *Spo11* KO *Fignl1* cKO, *Spo11*$^{-/-}$ *Fignl1*$^{flox/\Delta}$ *Stra8-Cre*$^+$. The results of the two-tailed Mann–Whitney *U*-test are indicated in the graphs: ****$p \le 0.0001$. The total number of cells analyzed is indicated below the graphs. Scale bars in (**a**), (**c**) and (**e**), 10 μm and 2 μm for magnified images in (**e**). Prelept., preleptonema; lept., leptonema; zyg., zygonema. Source data are provided as a Source Data file.

cKO testes were common to *Spo11* KO testes (94.1%; 241/256; Supplementary Fig. 10e), indicating a similar binding profile of RAD51 on ssDNAs between *Spo11* KO and *Spo11* KO *Fignl1* cKO mice. These suggest that, in the absence of FIGNL1, RAD51 does not bind to any specific regions with ssDNAs in *Spo11* KO spermatocytes.

To address whether RAD51 persists on meiotic recombination sites in the absence of FIGNL1, we also measured the colocalization frequency of RAD51 and RPA2 in mid/late zygonema, given that RPA2 localizes to recombination sites even after RAD51/DMC1 disassembly in wild-type meiosis and can be a marker of recombination sites on mid/late zygotene chromosomes. In control mid/late zygotene spermatocytes, out of 234 ± 44 RPA2 foci, 41 ± 15 foci were colocalized with RAD51 foci, which corresponds to 17.2 ± 4.3% of RPA2 foci (Fig. 6c–e). *Fignl1* cKO exhibited two classes of populations; one is similar to controls (~30%) and the other shows 40–80% of colocalization, with 45.4 ± 21.9% of RPA2 foci colocalized with RAD51 for all populations. This is consistent with the idea that RAD51 binds to ssDNA on meiotic recombination sites in the absence of FIGNL1, suggesting that RAD51 persists on RPA2-associated recombination sites.

### *Fignl1* deficiency rescues RAD51/DMC1- assembly defects in *Swsap1*

Our previous study in human cells showed that FIGNL1 antagonizes the function of the RAD51 mediator, SWSAP1[29]. In mouse male meiosis, SWSAP1 is necessary for efficient RAD51/DMC1- focus formation[11,29]. We tested the effect of *Fignl1* cKO on RAD51/DMC1 assembly in *Swsap1* KO cells by generating *Swsap1* KO *Fignl1* cKO male mice (*Swasp1*$^{-/-}$ *Fignl1*$^{flox/\Delta}$ *Stra8-Cre*$^+$). As reported[11,29], surface-spread chromosomes from *Swasp1* KO spermatocytes (*Swsap1*$^{-/-}$ *Fignl1*$^{flox/\Delta}$) showed a large reduction of RAD51 foci with 10-30 foci (~1/10 of control *Fignl1*$^{flox/\Delta}$) in leptonema and zygonema (29 ± 26, 22 ± 15, and 20 ± 14 foci in leptonema, early zygonema, and mid zygonema, respectively; Fig. 7a, b). Like *Fignl1* cKO, *Swsap1* KO *Fignl1* cKO spermatocytes showed an accumulation of RAD51 foci on meiotic chromosomes. The number of RAD51 foci from leptonema to mid-zygonema in *Swsap1* KO *Fignl1* cKO is very similar to that in *Fignl1* cKO (320 ± 217, 308 ± 221, and 194 ± 179 foci in *Swsap1* KO *Fignl1* cKO and 399 ± 145, 377 ± 98, and 282 ± 91 foci in *Fignl1* cKO in leptonema, early zygonema, and mid zygonema, respectively). This indicates that in mouse spermatogenesis without FIGNL1, RAD51 assembly does not depend on SWSAP1. This is consistent with the results in human cultured cells. On the other hand, in late zygonema, the number of RAD51 foci in *Swsap1* KO *Fignl1* cKO was comparable to *Swsap1* KO and much lower than that in *Fignl1* cKO (10 ± 12, 7 ± 8, and 189 ± 79 foci in *Swsap1* KO *Fignl1* cKO, *Swsap1* KO, and *Fignl1* cKO, respectively; Fig. 7b). This suggests that only *Swsap1*$^{-/-}$ *Fignl1*$^{flox/\Delta}$ cells where Cre recombinase failed excision of the *Fignl1*$^{flox}$ allele can progress into late zygotene stage in *Swsap1* KO *Fignl1* cKO testes. Moreover, similar results were observed for DMC1; the number of DMC1 foci in *Swsap1* KO *Fignl1* cKO was comparable to *Fignl1* cKO (Fig. 7c, d). Thus, reduced DMC1-focus formation in *Swsap1* KO is also suppressed by *Fignl1* cKO. It is likely that the activity of FIGNL1 to suppress RAD51/DMC1 assembly on intact dsDNAs is independent of SWSAP1-mediated stable assembly of RAD51/DMC1 on DSB-induced ssDNAs.

### FIGNL1 disrupts RAD51-dsDNA complex in vitro

We previously showed that FIGNL1 promotes the dissociation of RAD51 from ssDNAs in vitro[29]. The above results in mouse spermatocytes suggest that FIGNL1 can disrupt RAD51 filaments on chromosomal DNAs, both ssDNAs and dsDNAs. To address whether FIGNL1 promotes the dissociation of RAD51 filaments on dsDNAs in vitro, we purified human FIGNL1ΔN, which lacks N-terminal 284 amino acids[29], since a full-length human FIGNL1 protein was insoluble when expressed in *E. coli*, and performed the electrophoretic mobility shift assay (EMSA). First, we confirmed that a purified human FIGNL1ΔN disrupted the pre-formed RAD51-ssDNA complexes in the presence of ATP (Fig. 8a, b). Under this condition, FIGNL1ΔN could disrupt ~30% of RAD51-ssDNA complex (24.6 ± 6.9% of unbound DNAs were increased to 42.9 ± 2.1% by the addition of 1.2 μM of FIGNL1ΔN). When FIGNL1ΔN was incubated with a pre-formed RAD51-dsDNA complex, the most of dsDNA was recovered as an unbound form (25.8 ± 11.1% of unbound DNAs were increased to 80.0 ± 11.6%). This suggests that FIGNL1ΔN could disrupt the RAD51-dsDNA complex more efficiently than the RAD51-ssDNA complex in vitro, supporting the in vivo results. Alternatively, RAD51-ssDNA in vitro might form an aggregate more than RAD51-dsDNA, which may affect the accessibility by FIGNL1 protein to disrupt the complexes.

## Discussion

In this study, we analyzed the role of FIGNL1 AAA+ ATPase, which plays a critical role in homologous recombination in somatic cells, in mouse male meiosis. *Fignl1* cKO mice displayed higher levels of RAD51 and DMC1 foci than controls, as well as defective chromosome synapsis. Our analysis revealed the dual function of FIGNL1 during spermatogenesis (Fig. 8c). One is to promote meiotic recombination by regulating the dissociation of RAD51/DMC1 ensembles on the ssDNA. The other is to protect the genome by disrupting the non-functional form of RAD51/DMC1 ensembles on the dsDNA. Consistent with the idea, purified FIGNL1 can disrupt the RAD51-dsDNA complex as well as the RAD51-ssDNA complex in vitro. The same conclusion was obtained by Baudat's group by characterizing the *Fignl1* cKO as well as *Firrm* cKO[39].

### FIGNL1 promotes the post-assembly step of RAD51/DMC1 in meiotic recombination

In meiotic recombination, RAD51/DMC1 filaments on ssDNAs play a critical role in homology search and strand exchange with homologous dsDNA, leading to the formation of the D-loop, in which the 3'-OH end of the invading strand becomes a substrate for recombination-associated DNA synthesis. Biochemical analyses showed that RAD51/DMC1-mediated D-loop is a poor substrate for in vitro DNA synthesis by DNA polymerase δ and its accessory factors[59,60]. To promote efficient DNA synthesis on the recombination intermediate, RAD51/DMC1 bound

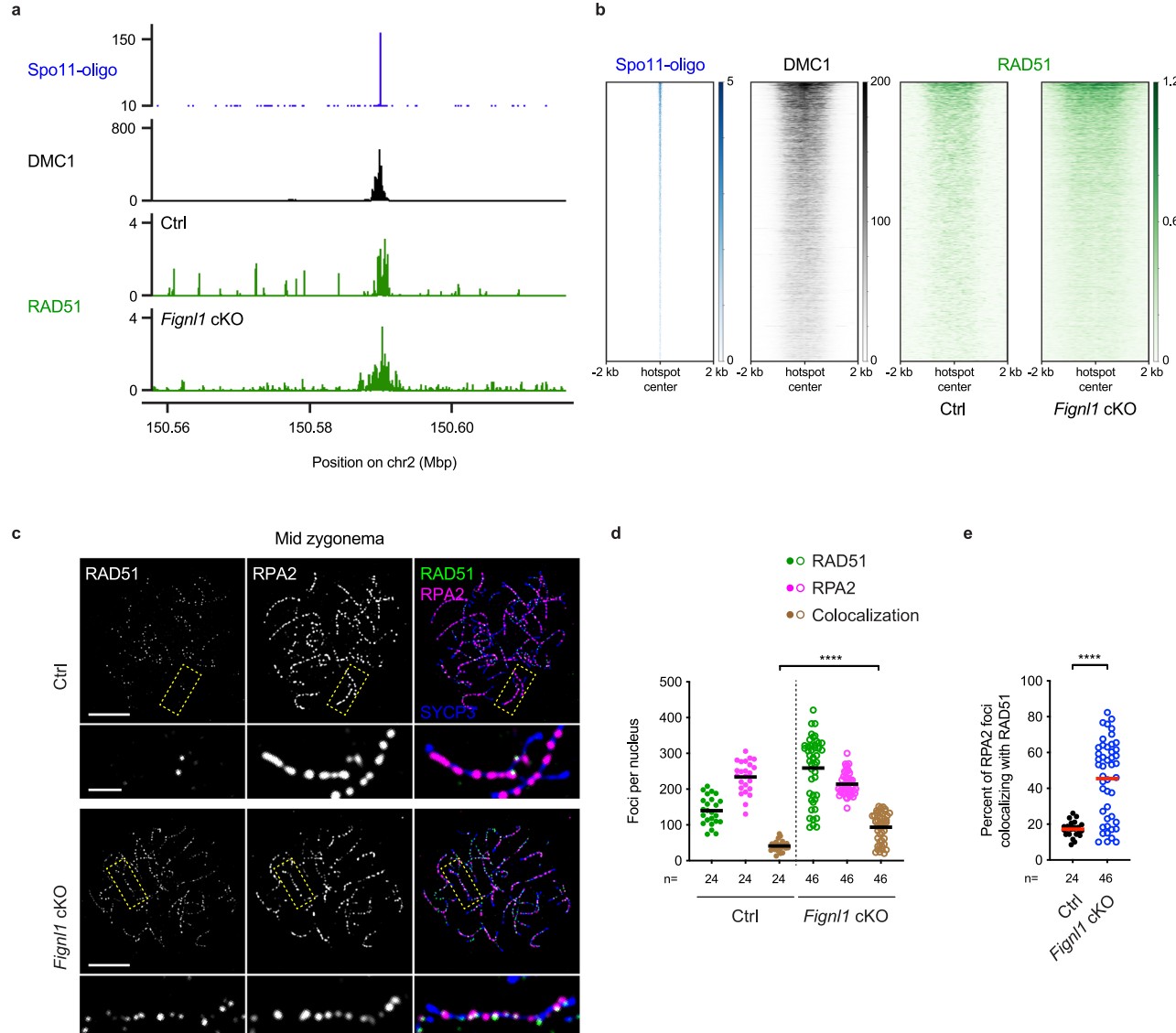

**Fig. 6 | Persistent RAD51 binding at recombination sites in *Fignl1* cKO spermatocytes. a** RAD51 ChIP-SSDS signals (green) in Ctrl and *Fignl1* cKO with Spo11-oligo counts (blue) and DMC1 ChIP-SSDS signals (black) in wild-type at an indicated chromosomal region with a hotspot on mouse chromosome 2. **b** The heatmap of RAD51 ChIP-SSDS signals (green) in Ctrl and *Fignl1* cKO and Spo11-oligo counts (blue) and DMC1 ChIP-SSDS signals (black) in wild-type. The 4000 most active meiotic recombination hotspots on autosomes identified by Spo11-oligo in wild-type are ordered by SPO11-oligos counts[77]. ChIP-SSDS signals are centered relative to hotspot centers. In (**a**) and (**b**), Spo11-oligo and DMC1 ChIP-SSDS data are from previous studies[77,78]. **c** Representative images of spermatocyte-chromosome spreads immunostained for RAD51 (green), RPA2 (magenta), and SYCP3 (blue) at midzygonema in Ctrl and *Fignl1* cKO. The bottom panels are magnified images of

regions with yellow dotted rectangles. Scale bars, 10 μm for whole-nucleus images and 2 μm for magnified images. **d** Quantification of focus numbers of RAD51 (green), RPA2 (magenta) and RAD51-RPA2 colocalization (brawn) at mid zygonema in Ctrl (filled circles) and *Fignl1* cKO (open circles). The black bars are means. **e** Quantification of the frequency of RAD51-RPA2 colocalization at midzygonema in Ctrl (black circles) and *Fignl1* cKO (blue open circles). The degree of RPA2 foci colocalizing with RAD51 is shown. The red bars are means. Genotypes of indicated animals are: Ctrl, *Fignl1^{+/+} Stra8-Cre^+*; *Fignl1* cKO, *Fignl1^{flox/Δ} Stra8-Cre^+*. The results of the two-tailed Mann−Whitney *U*-test are indicated: ****$p \leq 0.0001$. The total number of cells analyzed is indicated below the graphs. Source data are provided as a Source Data file.

to the intermediate should be removed by factors such as RAD54, a RAD51 remodeler[59,60]. Although the male KO mice for *Rad54* and *Rad54b*, a *Rad54* paralog, and the double KO mice for *Rad54* and *Rad54b* are fertile, the *Rad54 Rad54b* double KO spermatocytes show abnormal loading of RAD51 on chromosomes in pachynema and diplonema[32]. This is consistent with the role of RAD54 in the post-assembly step of RAD51 (DMC1 had not been analyzed in detail). Our in vivo and in vitro studies suggest the role of FIGNL1 in the post-assembly step of the RAD51/DMC1 filaments. It is likely that FIGNL1 disrupts not only RAD51 but also DMC1 filaments on recombination intermediates for subsequent DNA synthesis[59,61] (Fig. 8c), though other possibilities including that proper

disassembly of RAD51 and DMC1 filaments promotes strand invasion won't be excluded. Post-assembly role of FIGNL1 in RAD51-mediated homologous recombination was reported previously in human cells with DNA damage[34]. Consistent with the post-assembly role of FIGNL1, *Fignl1* cKO spermatocytes reduce the formation of MSH4 foci, which appears after the loading and dissociation of RAD51/DMC1[48]. *Fignl1* cKO spermatocytes seem not to enter the pachytene stage for the loading of the MutLγ complex such as MLH1 and most *Fignl1* KO cells die around late zygonema. The meiotic defects seen in *Fignl1* cKO are similar to those in the mutants where RAD51/DMC1 foci accumulate in a higher number than normal such as *Msh4*, *Msh5*, *Mcm8*, *Trip13*, and *Mcmdc2*

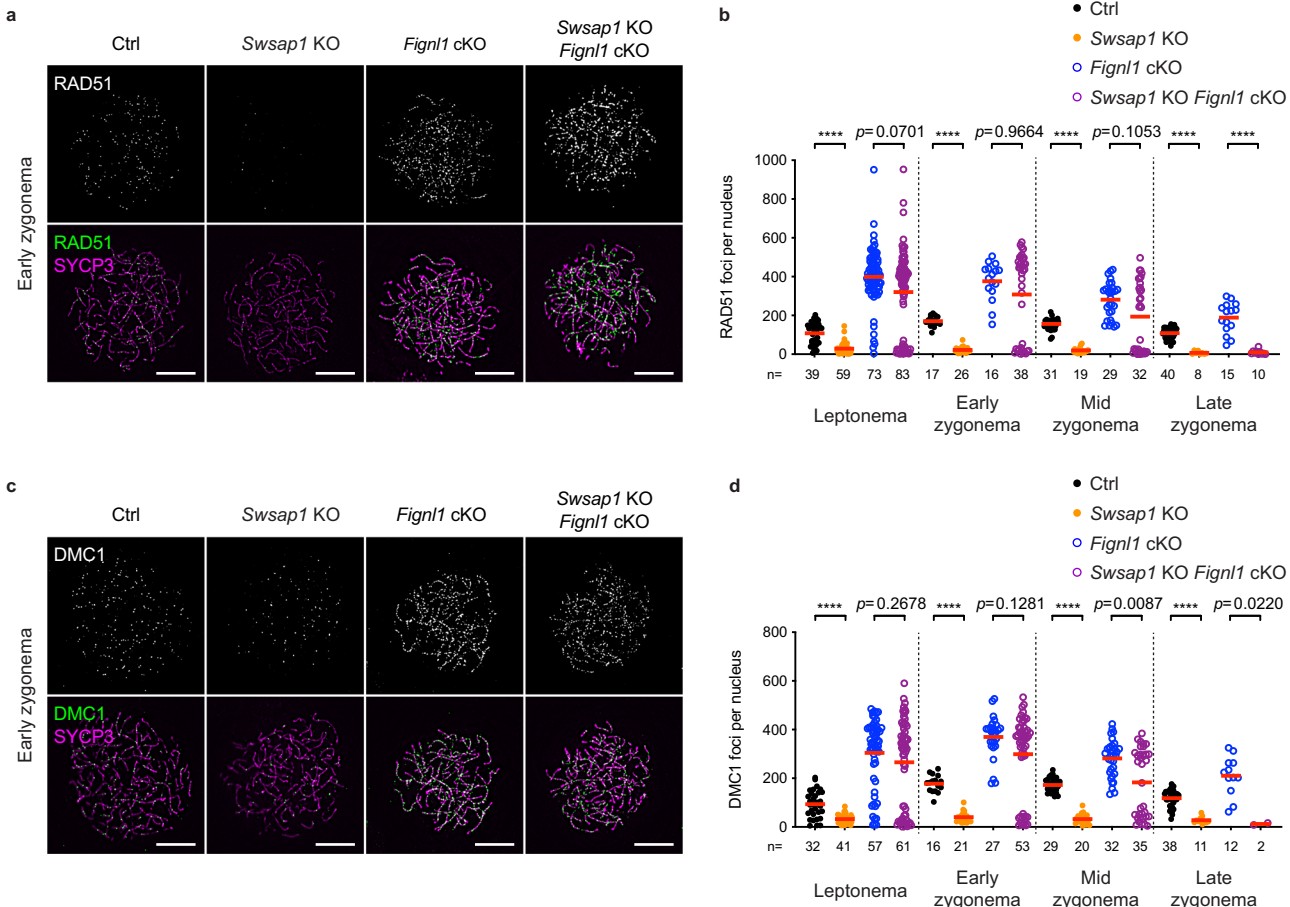

**Fig. 7 | *Fignl1* deficiency rescues defects in RAD51 and DMC1 assembly caused by *Swsap1* deficiency. a** Representative images of spermatocyte-chromosome spreads immunostained for RAD51 (white in the top panels and green in the bottom panels) and SYCP3 (magenta) at early zygonema in Ctrl, *Swsap1* KO, *Fignl1* cKO and *Swsap1* KO *Fignl1* cKO. **b** Quantification of RAD51 focus numbers at different meiotic prophase I stages in Ctrl (black circles), *Swsap1* KO (orange circles), *Fignl1* cKO (blue open circles), and *Swsap1* KO *Fignl1* cKO (purple open circles). The red bars are means. **c** Representative images of spermatocyte-chromosome spreads immunostained for DMC1 (white in the top panels and green in the bottom panels)

and SYCP3 (magenta) at early zygonema in Ctrl, *Swsap1* KO, *Fignl1* cKO, and *Swsap1* KO *Fignl1* cKO. **d** Quantification of DMC1 focus numbers at different meiotic prophase I stages in Ctrl (black circles), *Swsap1* KO (orange circles), *Fignl1* cKO (blue open circles), and *Swsap1* KO *Fignl1* cKO (purple open circles). The red bars are means. Genotypes of indicated animals are: *Swsap1* KO, *Swsap1$^{-/-}$ Fignl1$^{flox/\Delta}$*; *Fignl1* cKO, *Fignl1$^{flox/\Delta}$ Stra8-Cre$^+$*; *Swsap1* KO *Fignl1* cKO, *Swsap1$^{-/-}$ Fignl1$^{flox/\Delta}$ Stra8-Cre$^+$*. The results of the two-tailed Mann-Whitney *U*-test are indicated in the graphs: ****$p \le 0.0001$. The total number of cells analyzed is indicated below the graphs. Scale bars in (**a**) and (**c**), 10 μm. Source data are provided as a Source Data file.

mutant spermatocytes[62,63]. In mouse spermatocytes, FIGNL1 seems to promote ZMM-dependent CO formation. This is different from the role of FIGNL1 ortholog, FIGL1, in *A. thaliana*. *Arabidopsis* FIGL1 mutant meiocytes showed normal or slightly elevated CO formation[35,36]. Indeed, the depletion of FIGL1 or its partner FLIP suppresses reduced CO formation in the absence of pro-CO proteins, ZMM[35,36]. This suggests the role of plant FIGL1-FLIP as an anti-CO factor, which disrupts early recombination intermediates such as RAD51/DMC1 filaments on the ssDNA prior to or during the strand invasion. Although our attempt to analyze the role of FIGNL1 in mouse oocytes failed due to insufficient excision of the *Fignl1$^{flox}$* allele in oocytes by the *Stra8-Cre* transgene used in this study, a recent study by Yu and colleagues demonstrated the accumulation of RAD51 in not only *Firrm/Flip* cKO spermatocytes but also in oocytes with infertility in both male and female mice[55], suggesting that the essential role of the FIGNL1-FIRRM complex in meiotic recombination is shared between males and females.

**FIGNL1 disrupts non-productive RAD51 and DMC1 ensembles on duplex DNAs**
RAD51/DMC1 focus formation normally occurs at sites of ssDNA tracks that are associated with mitotic or meiotic DSBs. The most striking phenotypes seen in *Fignl1* cKO spermatocytes are the accumulation of

RAD51 and DMC1 foci in unchallenged pre-meiotic S-phase cells and spermatocytes in the absence of SPO11-dependent meiotic DSBs. This accumulation of RAD51/DMC1 foci in the absence of DSBs, therefore the ssDNAs, could be simply explained by the non-specific assembly of RAD51/DMC1 filaments on chromatin DNAs, thus the binding to dsDNA (Fig. 8c). Interestingly, human RAD51 and DMC1 bind to both ssDNAs and dsDNA with similar affinity in vitro[10,64] and RAD51 bears an activity to remodel the nucleosome upon the dsDNA binding[65,66]. On the other hand, detection of RAD51/DMC1 foci in vivo normally requires the presence of tracts of ssDNAs. How cells discriminate ssDNAs among lots of competitors, dsDNAs, in the RAD51/DMC1 binding remains unsolved. Based on the results described here, we propose that FIGNL1 actively dismantles RAD51 and DMC1 filaments on dsDNAs, which is otherwise non-productive for the recombination and/or might become obstacles to chromosomal events on duplex DNAs such as DNA replication, transcription, and others. Indeed, Bishop, Heyer and their colleagues reported activities of the RAD54 translocase family to remove RAD51 and DMC1 from dsDNAs. In yeast meiosis, Spo11-independent focus formation of Dmc1, but not of Rad51, depends on Tid1/Rdh54, a Rad54 homolog[67]. Mitotic spontaneous Rad51-focus formation on yeast chromosomes is increased in the absence of three translocases, Rad54, Tid1/Rdh54, and Uls1[68].

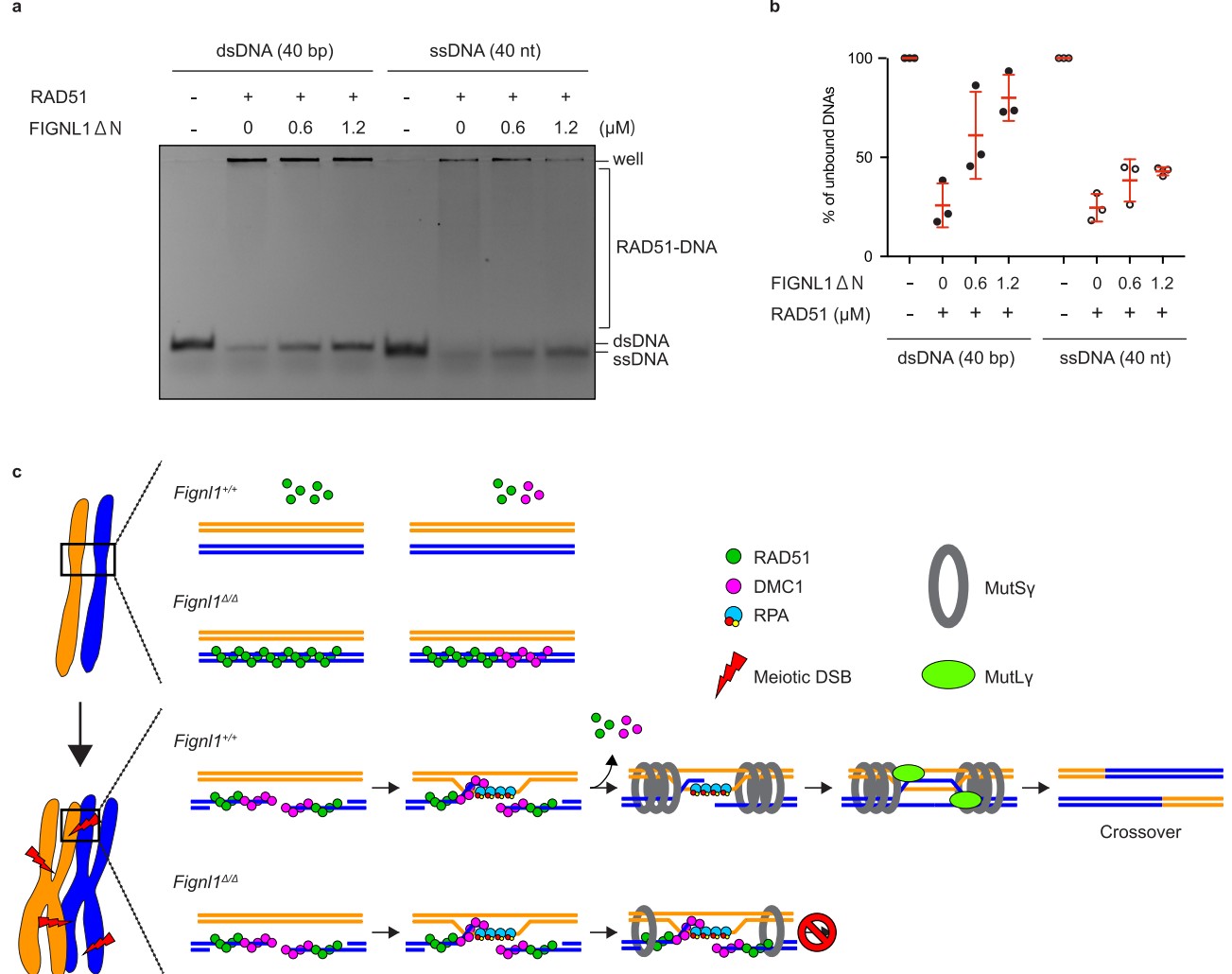

**Fig. 8 | FIGNL1 disrupts RAD51-DNA complexes. a** Electrophoresis mobility shift assay showing in vitro disruption of RAD51-DNA complexes by FIGNL1. A representative gel image among triplicate is shown. Human RAD51 protein was incubated with either Cy5-labeled ssDNA (40nt) or dsDNA (40 bp) in the presence of ATP. After the formation of the protein-DNA complex, purified human FIGNL1ΔN was added at the indicated concentration and further incubated. The complexes were analyzed by polyacrylamide gel electrophoresis and the fluorescence from the DNAs was captured. **b** Quantification of free ssDNA or dsDNA in (**a**). The red bars are means and SDs from three independent experiments. **c** Model of FIGNL1 function in spermatocytes. In pre-meiotic S-phase and early meiotic prophase I, FIGNL1 dissociates RAD51 and DMC1 from dsDNAs. At meiotic DSB sites, FIGNL1 removes RAD51 and DMC1 from ssDNAs, which facilitates efficient loading of MutSγ (MSH4-MSH5) and further processing of homologous recombination to crossover outcome. Source data are provided as a Source Data file.

Moreover, in unchallenged human cells, the absence of two RAD54 family proteins RAD54 (or RAD54L) and RAD54B leads to the accumulation of spontaneous RAD51 foci particularly when RAD51 is over-expressed[69]. These in vivo studies were further supported by biochemical analysis showing that Rad54 dissociates RAD51 from dsDNAs[70,71]. Further studies are necessary to reveal the relationship between FIGNL1 and RAD54 translocases in the disassembling of RAD51/DMC1 on duplex DNAs. We stress that, distinct from RAD54, which dismantles RAD51 filaments only on dsDNAs (not on ssDNAs), FIGNL1 can disrupt RAD51 ensembles on both ss- and dsDNAs.

One of the striking features of RAD51/DMC1 assembles on duplex DNAs is that these ensembles are quite stable. The number of RAD51 in both pre-meiotic S-phase and early meiotic prophase I from early leptonema to early zygonema is constant. Little increase in RAD51-focus number during early meiotic prophase I suggests that spermatocytes show very limited additional assembly of RAD51 foci during this stage. On the other hand, DMC1 shows a two-fold increase in focus number from early leptonema (and pre-meiotic S-phase) to early zygonema. This increase could be induced by SPO11-mediated DSBs.

Interestingly, the colocalization frequency of RAD51 with DMC1 in *Fignl1* cKO increased from ~30% to ~60%, which is similar to the colocalization frequency seen in control early zygonema, while the colocalization frequency of DMC1 with RAD51 is nearly constant (with 60–70% frequency). This suggests that RAD51 ensembles even on the dsDNA tend to recruit DMC1 to form co-ensembles.

## FIGNL1 antagonizes SWSAP1 for RAD51/DMC1 assembly on meiotic chromosomes

FIGNL1 shows antagonism to the RAD51 mediator. In human cells, *FIGNL1* depletion suppresses RAD51-assembly defect induced by the *SWSAP1* depletion[29]. In meiosis of the *A. thaliana*, *FIGL1* (*A. thaliana FIGNL1*) and/or *FLIP* deletion restores reduced RAD51/DMC1-focus formation in the mutants of *BRCA2*[38] as well as defective DMC1-focus formation in the mutant of the *SDS1*, which promotes DMC1 assembly[35]. Since *Swsap1* KO spermatocytes reduced the RAD51 and DMC1 foci[11,29], we asked whether *Fignl1* cKO could also suppress these meiotic defects in *Swsap1* KO spermatocytes and found that the number of both RAD51/DMC1 foci in *Swsap1* KO *Fignl1* cKO is similar to

those in *Fignl1* cKO. This is simply interpreted as, like in human cells, *Fignl1* cKO suppresses RAD51/DMC1-assembly defect in *Swsap1* KO. However, at this point, we do not know whether RAD51/DMC1 foci are associated with ssDNAs in meiotic prophase I cells of *Swsap1* KO *Fignl1* cKO as shown in *Fignl1* cKO, although it is likely. In addition, different from RAD51/DMC1 assembly on ssDNAs, these suggest that RAD51/DMC1 assembly on the dsDNA in the absence of FIGNL1 does not require the RAD51/DMC1 mediator, SWSAP1.

## Methods

### Mice
The care and use of mice in this study were performed under the guideline for the proper conduct of animal experiments (Society Council of Japan). These procedures were approved by the Institutional Animal Care Committee at Institute for Protein Research, Osaka University (approval ID; 25-03-0). Mice were housed with food and water provided *ad libitum* and maintained in a temperature-controlled room at 22 °C on a 12 h light:12 h dark cycle.

All mice used in this study were congenic to the C57BL/6 background. Genotyping of mice was performed by PCR on genomic DNA isolated from mice. Primers used for genotyping are listed in Supplementary Table 1. Mature adult mice (8–20 weeks old) were used for experiments otherwise mentioned.

*Fignl1^flox^* allele was constructed by Cyagen Bioscience Co. Ltd. ES cells with the *Fignl1^flox^* allele were injected into C57BL/6 albino blastocytes, which were transplanted into CD-1 pseudo-pregnant female mice. The founder mice with the correct *Fignl1^flox^* allele were backcrossed to C57BL/6 mice. *Fignl1^+/Δ^* allele was constructed by breeding *Fignl1^flox/+^* with *CAG-Cre* mice. *Stra8-Cre* transgenic mice were previously described[72] and provided by the RIKEN Animal Resource Center. *Spo11^-/-^* mouse was previously described[53] and *Spo11^+/-^* mice with a mixed background of C57BL/6 and 129/SvJ were backcrossed to C57BL/6 mice prior to experiments. *Swsap1^-/-^* mice were described in our previous study[29].

### Histology
Testes from adult mice were fixed in 10% buffered formalin (Wako, 062-01661) overnight at room temperature, washed twice with PBS, and stored in 70% EtOH at 4 °C prior to embedding in paraffin and sectioning. The sections of 5 μm thickness were deparaffinized and stained with periodic acid-Schiff (PAS).

### Sperm count
A pair of Cauda epididymis from adult mice (11–17 weeks) were placed in 1 ml of D-MEM (Wako, 044-29765)/10% bovine serum albumin (BSA) in a 35 mm petri dish (Corning, 430165) and torn to pieces using two fine forceps for 3 min. After incubation at 32 °C for 20 min, 20 μl of sperm suspension was fixed in 480 μl 10% buffered formalin (Wako, 062-01661) at room temperature for counting.

### Spermatocyte-chromosome spreads
Surface-spread chromosomes of spermatocytes were prepared as described previously[73] with slight modification. Testes were dissected, the tunica albuginea was removed, and adherent extra-tubular tissues were removed by rinsing and dissociating seminiferous tubules in PBS using a pair of 25-gauge needles in a 35 mm petri dish (Corning, 430165). Dissociated seminiferous tubules were incubated in hypotonic extraction buffer (30 mM Tris-HCl pH 8.0, 50 mM sucrose, 17 mM trisodium citrate dihydrate, 5 mM EDTA, 0.5 mM Dithiothreitol (DTT) and 0.5 mM phenylmethylsulphonyl fluoride (PMSF), pH 8.2–8.3; HEB) for 15–45 min at room temperature. After quickly rinsing seminiferous tubules in 100 mM sucrose, some tubules were placed in 40 μl of 100 mM sucrose on a glass depression slide. Tubules were torn to pieces using two fine forceps, and pieces of tubular remnants were removed. The cell suspension was made by pipetting, and 20 μl of cell

suspension was placed on a clean glass slide (Matsunami, MAS-01) coated with freshly made 1% paraformaldehyde (PFA) solution pH 9.2 (pH set by 1.25 M sodium borate) containing 0.15% Triton X-100. Slides were placed and dried in a closed humid chamber with hot tap water overnight at room temperature, followed by drying with lids ajar for 3 h and then with lids removed for 1–2 h. Slides were washed once for 5 min in deionized water and twice for 5 min in 0.4% DryWell (Fujifilm) in a coplin jar and air-dried at room temperature. Slides were either directly processed for immunostaining as below or stored in aluminum foil at −80 °C prior to immunostaining.

For the preparation of testicular cell suspension, testes were dissected, the tunica albuginea was removed, and seminiferous tubules were transferred into 1 ml of testis cell isolation medium (104 mM NaCl, 45 mM KCl, 1.2 mM MgSO$_4$, 0.6 mM NaHPO$_4$, 0.15 (w/v) glucose; TIM) containing 2 mg of collagenase (Worthington, LS004186) for incubation at 32 °C for 55 min with shaking at 53 rpm. Tubules were washed three times with 3 ml of TIM, and suspended in 1 ml of TIM containing 0.7 mg of Trypsin (Worthington, LS003740) and 4 μg of DNase I (Roche, 104159) for incubation at 32 °C for 15 min with shaking at 53 rpm. Trypsinization was stopped by adding 5 mg of Trypsin inhibitor (Sigma, T9003) and 10 μg of DNase I, and cells were separated by pipetting and filtered through a 70 μm cell strainer (Corning, 431751). Cells were washed once with 0.75 μg of DNase I and 7.5 ml of TIM, and suspended in 0.75 μg of DNase I and PBS (Nakarai, 14249-24) to $1 \times 10^6$ cell/ml. 1 ml of cell suspension was centrifuged at $200 \times g$ for 5 min at room temperature. After carefully removing the supernatant, cells were suspended in 30 μl of HEB and incubated for 15–45 min at room temperature. 30 μl of 100 mM sucrose was added, and 20 μl of cell suspension was placed on a clean glass slide (Matsunami, MAS-01) coated with 100 μl of freshly made 1% paraformaldehyde (PFA) solution pH 9.2 containing 0.15% Triton X-100. Slides were placed and dried in a closed humid chamber, washed, and stored as described above.

### Immunofluorescence staining
Slides were rehydrated with Tris-buffered saline (TBS) pH 8.0 containing 0.05% of Triton X-100 (TBST) for 3 min, blocked twice with blocking buffer (1% normal goat or donkey serum, 3% bovine serum albumin (BSA), 1x TBS pH 8.0, 0.05% of Triton X-100, 0.05% sodium azide) for 15 min at room temperature and incubated with primary antibodies in antibody dilution buffer (10% normal goat or donkey serum, 3% BSA, 1x TBS pH 8.0, 0.05% of Triton X-100, 0.05% sodium azide) in a humid chamber overnight at room temperature. Slides were quickly rinsed with TBST, washed twice with TBST for 5 min, blocked twice with blocking buffer for 15 min at room temperature, and incubated with secondary antibodies in antibody dilution buffer in a humid chamber for 1 h at room temperature. Slides were quickly rinsed with TBST, washed three times with TBST for 5 min, washed once with Milli-Q water for 2 min, and air-dried before mounting with ProLong Diamond Antifade Mountant (Thermo Fisher Scientific, P36970). For detection of IdU, rehydrated slides were denatured with 4 N HCl for 10 min at room temperature before blocking. EdU was detected using Click-iT EdU Imaging Kit with Alexa Fluor 647 azides according to the manufacturer's instruction (Invitrogen). All primary and secondary antibodies used are listed in Supplementary Table 2.

### Testicular cell squashes
Testes were dissected, the tunica albuginea was removed, and adherent extra-tubular tissues were removed by rinsing and dissociating seminiferous tubules in PBS using a pair of 25-gauge needles in a 35 mm petri dish (Corning, 430165). Dissociated seminiferous tubules were fixed in freshly prepared 2% formaldehyde containing 0.1% Triton X-100 for 10 min at room temperature. Small pieces of tubules were placed on a clean glass slide (Matsunami, MAS-01), torn using two fine forceps, and squashed under a coverslip with pressure from the blunt

end of pencil followed by a thumb. Slides were frozen briefly in liquid nitrogen and either directly processed for immunostaining or stored at −80 °C prior to immunostaining. After the removal of coverslips, slides were washed three times with PBS for 5 min and immunostaining was performed as described above.

## Image acquisition

Images of mouse spermatocyte spreads and testicular cell squashes were acquired using a computer-assisted fluorescence microscope system (DeltaVision; Applied Precision) with a 60× NA 1.4 oil immersion objective. Image deconvolution was performed using an image workstation (softWoRx version 6.5.2; Applied Precision) and afterward processed using Imaris (version 9.2.1; Oxford Instruments, UK) and Photoshop (version 23.2.2; Adobe, USA) software tools. Images in each figure are matched exposure.

## Image analysis

Comparisons were made between animals that were either littermate or matched by age. Single cells were manually cropped, and numbers of foci were counted by the auto-thresholding signal intensity in Imaris software (version 9.2.1; Oxford Instrument) and clear foci that were missed by auto-thresholding, especially adjacent foci, were manually counted. Signal intensities were measured from non-deconvolved images using Image J software (Fiji version 2.9.0) by subtracting signal intensities of single cells from those of regions of interest (ROIs) manually drawn next to cells as backgrounds.

Meiotic prophase I stages on spermatocyte spreads were defined by SYCP3 and SYCP1 staining using standard criteria. Leptonema was defined by short SYCP3 stretches without evidence of synapsis determined by SYCP1 staining. Zygonema was defined by longer stretches of SYCP3 with various degrees of synapsis: early, mid- and late zygonema was defined by having <25%, 25–75%, and >75% of synapsis, respectively. Pachynema was defined by the full synapsis of all autosomes. Diplonema was defined by de-synapsis beginning from mostly the middle of chromosomes with various degrees of residual synapsis. Preleptonema was defined by very short SYCP3 stretches with EdU or IdU-positive staining.

## EdU incorporation

1 ml of $1 \times 10^6$ cell/ml testicular cell suspension in PBS was centrifuged at $200 \times g$ for 5 min at room temperature, and after removal of the supernatant, cells were suspended in 500 µl of D-MEM (Wako, 044-29765)/10% FBS containing 10 µM EdU. After incubation at 37 °C for 10 min or 1 h in a well of a 12-well dish (Violamo, 2-8588-02), cells were pelleted by centrifugation at $200 \times g$ for 5 min at room temperature, washed in 1 ml of PBS once, and suspended in 30 µl HEB for incubation for 15–30 min at room temperature. After adding 30 µl of 100 mM sucrose, cells were fixed, and the slides were dried, washed, and stored as described above.

## IdU pulse incorporation

1 ml of $1 \times 10^6$ cell/ml testicular cell suspension in PBS was centrifuged at $200 \times g$ for 5 min at room temperature, and after removal of the supernatant, cells were suspended in 500 µl of D-MEM (Wako, 044-29765) / 10% FBS containing 20 µM IdU (Sigma, I7125). After incubation at 37 °C for 10 min in a well of a 12-well dish (Violamo, 2-8588-02), cells were subjected to chromosome spreads as described above.

## RAD51 Chromatin immuno-precipitation – single-stranded DNA sequencing (ChIP-SSDS)

ChIP was performed as described previously[56]. One testis from the control and a pair of testes from *Fignl1* cKO, *Spo11* KO, and *Spo11* KO *Fignl1* cKO mice (9–20 weeks, except a *Spo11* KO mouse at 31 weeks) were dissected, the tunica albuginea was removed, and seminiferous tubules were crosslinked in 10 ml of 1% formaldehyde (Pierce, 28908)

in PBS (Nakarai, 14249-24) for 10 min at room temperature. Cross-linking was quenched by adding 2.5 M glycine to a final concentration of 125 mM. After 5 min of incubation at room temperature, cells were washed once with 10 ml of cold PBS, and the cell pellet was snap-frozen on liquid nitrogen and stored at −80 °C. The frozen cell pellet was suspended in 500 µl of cell lysis buffer (0.25% Triton X-100, 10 mM EDTA, 0.5 mM EGTA, 10 mM Tris-HCl pH 8.0) supplemented with protease inhibitor (Roche, 04693159001), incubated for 10 min on ice, and centrifuged at $300 \times g$ for 3 min at 4 °C. The nuclei pellet was washed in 500 µl of lysis wash buffer (200 mM NaCl, 1 mM EDTA, 0.5 mM EGTA, 10 mM Tris-HCl pH 8.0) supplemented with protease inhibitor, resuspended in 200 µl of RIPA buffer (10 mM Tris-HCl pH 8.0, 1 mM EDTA, 0.5 mM EGTA, 1% Triton X-100, 0.1% sodium deoxycholate, 0.1% SDS, 140 mM NaCl) supplemented with protease inhibitor, split into two ~130 µl aliquots and sonicated on the Covaris M220 for 5 min 30 sec at 5–9 °C with a setting of Peak Power 50.0, Duty Factor 20.0, and Cycles/Burst 200. Sheared nuclei were combined, and after adding 700 µl of RIPA buffer, centrifuged at $20{,}400 \times g$ for 15 min at 4 °C to remove insoluble debris. 800 µl of the chromatin-containing supernatant was mixed with 50 µl of Dynabeads protein G bound to 9.7 µg of RAD51 antibody (Novus, NB100-148) and incubated overnight on a rotating wheel at 4 °C for immunoprecipitation. Beads were washed once with 1 ml of cold ChIP wash buffer 1 (0.1% SDS, 1% Triton X-100, 2 mM EDTA, 20 mM Tris-HCl pH 8.0, 150 mM NaCl), 1 ml of cold ChIP wash buffer 2 (0.1% SDS, 1% Triton X-100, 2 mM EDTA, 20 mM Tris-HCl pH 8.0, 500 mM NaCl), 1 ml of cold ChIP wash buffer 3 (0.25 M LiCl, 1% NP-40, 1% sodium deoxycholate, 1 mM EDTA, 10 mM Tris-HCl pH 8.0), and twice with 1 ml of cold TE (1 mM EDTA and 10 mM Tris-HCl pH 8.0), followed by elution with 100 µl of elution buffer (0.1 M NaHCO$_3$ and 1% SDS) for 30 min at 65 °C with vortexing every 10 min. The elutes were incubated overnight at 65 °C for de-crosslinking, neutralized by adding 4 µl of 1 M Tris-HCl (pH 6.5) and 2 µl of 0.5 M EDTA, and RNA was digested by adding 2 µl of 10 mg/ml RNase-A and incubation for 15 min 37 °C. After adding 5 µl of 10 mg/ml Proteinase-K (Wako, 165-21043) and incubation for 1 h at 55 °C, DNA was purified using QIAquick PCR Purification Kit (QIAGEN).

SSDS was performed as described previously[74]. 7–10 ng of ChIP DNA was further sonicated on the Covaris M220 for 8 min 30 sec at room temperature with a setting of Peak Power 75.0, Duty Factor 10.0, and Cycles/Burst 200, followed by end-repair and dA-tailing using ChIP-Seq Library Prep Reagent Set for Illumina (NEB, E6200L). DNA was incubated for 3 min at 95 °C and returned to room temperature for kinetic enrichment for single-stranded DNA. After adapter ligation using IDT for Illumina TruSeq DNA UD Indexes (Illumina, 20022370), SSDS libraries were amplified during 15 cycles of PCR using TruSeq DNA Nano LP (Illumina, 20016328), purified with AMPureXP beads (Beckman Coulter, A63880), and sequenced on HiSeq2500 in the rapid mode in a 50 bp PE run. Around 70 million paired reads were generated per sample.

A published bioinformatic pipeline (https://nf-co.re/ssds/dev) was run for the resulting reads for the identification of single-stranded sequences and mapping to the mm10 genome[74]. Heatmaps were generated using Deeptools[75].

## Western blot

Testes were dissected, the tunica albuginea was removed, and seminiferous tubules were snap frozen on liquid nitrogen and stored at −80 °C before cell lysis. Tubules from either one testis (*Fignl1*+/+ *Stra8-Cre*+ and *Fignl1*flox/Δ) or two testes (*Fignl1*flox/Δ *Stra8-Cre*+) were homogenized in 1 ml of RIPA buffer (50 mM Tris-HCl pH 7.5, 150 mM NaCl, 1 mM EDTA, 1% NP-40, 0.5% sodium deoxycholate, 0.1% sodium dodecyl sulfate (SDS)) supplemented with protease inhibitor (Roche, 04693159001) on ice, incubated for 15 min on ice, sonicated with a Bioruptor for 30 min (30 cycles of 30 s ON / 30 s OFF) at 4 °C with a setting of High intensity, and incubated for 15 min on ice. The samples

**Article**

were centrifuged at 17,800 × *g* for 15 min at 4 °C and the supernatants were collected as whole-testis extracts. Protein concentrations were measured by Bradford assay and the concentrations between samples were made constant by adding RIPA buffer supplemented with protease inhibitor. Whole-testis extract was mixed with 4x SDS-PAGE loading buffer (250 mM Tris-HCl pH 6.8, 8% SDS, 40% glycerol, 572 mM β-mercaptoethanol, 0.05% bromophenol blue) and boiled for 5 min at 100 °C and subjected to immunoblotting.

Protein samples were separated on 10% gels by SDS-PAGE and transferred to PVDF membranes by wet transfer method in Towbin buffer (25 mM Tris, 192 mM glycine, 20% ethanol) at 60 V for 150 min at 4 °C. Membranes were blocked with PBS containing 2.5% non-fat milk for 1 h at room temperature and incubated with primary antibodies in 2.5% non-fat milk in PBS overnight at 4 °C. Membranes were washed three times with PBS containing 0.05% Tween-20 (PBST) for 10 min and incubated with HRP-conjugated secondary antibodies in PBST for 1 h at room temperature. Membranes were washed three times with PBST for 10 min and the signal was developed by ECL Prime Western Blotting Detection Reagent (Amersham, RPN2232) and detected by Image-Quant LAS4000 (GE Healthcare).

### Antibody generation

His-tagged mouse SYCP3 was overexpressed from pET28c-SYCP3 plasmid by the addition of 0.2 mM IPTG in BL21(DE3) at 37 °C for 4 h. Cell extracts were prepared in Lysis buffer (50 mM Tris-HCl pH 7.5, 500 mM NaCl) by sonication and the supernatants were collected after centrifugation at 20,400 × *g* for 30 min at 4 °C. After adding Tween-20 to a final concentration of 0.2%, the cell lysates were incubated with Ni-NTA Agarose (QIAGEN) for 2 h at 4 °C and the lysate-resin slurry was washed in Lysis buffer containing 50 mM Imidazole, 100 mM Imidazole, 300 mM Imidazole. Ni-NTA-bound His-SYCP3 was eluted in Lysis buffer containing 3 M Imidazole and 3 M Urea (elution 1) by incubation for 1 h at 4 °C and in Lysis buffer containing 0.1% SDS (elution 2) by incubation for 30 min at room temperature. Elution 2 was concentrated using MICROCON Centrifugal Filter Device YM-10 (Millipore) and subjected to immunization to a rat.

### Purification of RAD51 and FIGNL1ΔN

GST-tagged human FIGNL1ΔN (N-terminal 284 aa deletion) was overexpressed from pGEX-6P-1-*Fignl1ΔN* plasmid by the addition of 0.25 mM IPTG in BL21(DE3) pLysS at 30 °C for overnight. Cell extracts were prepared in Lysis buffer (25 mM Tris-HCl pH 7.6, 150 mM NaCl, 10% Glycerol, 1 mM DTT) and GST-FIGNL1ΔN was recovered by GSTrap HP column (Cytiva). FIGNL1ΔN was eluted from the GSTrap column by the cleavage of GST tag by Prescission protease (#27-0843-01, Cytiva) in SP buffer (20 mM HEPES-NaOH pH 7.5, 10% Glycerol, 1 mM DTT) supplemented with 100 mM NaCl. FIGNL1ΔN was further purified by HiTrap SP column (Cytiva) with a linear gradient of 100 mM-1M in SP buffer. The peak fractions are dialyzed against SP buffer supplemented with 100 mM NaCl, concentrated with Amicon Ultra 30 K (Millipore), and stored at −80 °C. Human RAD51 was purified as described previously using pET-HsRad51 plasmid[76].

### Electrophoresis mobility shift assay (EMSA)

Double-stranded DNA substrate was prepared by annealing 5′-Cy5-labeled, 40-mer oligo DNA (Oligo 1, Cy5-TAATACAAAATAAGT AAATGAATAAACAGAGAAAATAAAG, IDT) and its complementary DNA (Oligo 2, CTTTATTTTCTCTGTTTATTCATTTACTTATTTTGTA TTA, Fasmac). Oligo 1 was used as an ssDNA substrate. For the assay, DNA (0.8 μM in bp/nt) was pre-incubated with 1 μM of RAD51 in 10 μl of the buffer (20 mM HEPES-NaOH pH7.5, 100 ug/ml BSA, 1 mM DTT, 10 mM MgCl$_2$, 2 mM ATP) at 37 °C for 5 min to form RAD51-DNA complex. Then the indicated concentration of FIGNL1ΔN was added to the reaction and further incubated at 37 °C for 15 min. In the control reaction without FIGNL1ΔN, the same volume of dialysis buffer was

added instead of the protein. DNA-protein complexes were analyzed using 8% native polyacrylamide gel electrophoresis with 1×TBE. Cy5-labeled DNA was visualized by LAS4000 (GE healthcare) and unbound DNA was quantified using ImageQuant TL (GE healthcare). Images were further processed to adjust brightness using Photoshop (Adobe, USA) software.

### Statistical analysis and reproducibility

Statistical analyses were performed using Graphpad Prism software version 9.5.1 and R version 4.2.2 (https://www.r-project.org). Bars in figures and values in manuscripts indicate means ± standard deviations (SDs) otherwise mentioned. Statistic parameters and tests and sample sizes are described in the figures and/or figure legends. Sample sizes were not predetermined using any statistical tests.

At least two animals of each genotype were analyzed and similar results were obtained otherwise mentioned. Three independent experiments were performed for biochemical analysis.

### Reporting summary

Further information on research design is available in the Nature Portfolio Reporting Summary linked to this article.

## Data availability

The raw and processed sequencing data generated in this study have been deposited in NCBI's Gene Expression Omnibus and are accessible through GEO Series accession number GSE227944. Uncropped immunoblotting images and an unprocessed image of electrophoresis mobility shift assay are provided in a Source Data file. The raw data used in plots and graphs are all provided in a Source Data file. The raw microscopy images used for figures and quantification are available from corresponding authors upon request. Source data are provided with this paper.

## Code availability

A published bioinformatic pipeline for single-stranded DNA sequencing analysis is available at https://nf-co.re/ssds/dev or https://github. com/nf-core/ssds/tree/dev.

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

## Acknowledgements

We thank Drs. F. Baudat and B. de Massy for sharing unpublished results. We acknowledge Drs. F. Pratto and V. Jayakumar for SSDS analysis. We are grateful to Ms. S. Kondo for the generation of anti-SYCP3. We acknowledge Drs. S. Keeney, M. Jasin, Y. Fujiwara, and K. Ishiguro for *Spo11* mice and Dr. H. Kurumizaka for the RAD51 expression plasmid. We are also indebted to members of the Shinohara lab, particularly Ms. S. Aoyama, A. Maeda, C. Watanabe, M. Yasumura, and S. Hashimoto for technical assitance. This work was supported by JSPS KAKENHI Grant Numbers; 19H00981 to A.S., 20K15716 and 16H06279 (PAGS) to M.I., and 19H03157 to A.F.

## Author contributions

M.I. and A.S. conceived and designed the experiments. M.I. performed all experiments on mouse meiosis. M.I., Y.F., and A.T. analyzed SSDS data. A.F. performed biochemical experiments. K.M. set up Fignl1 mice. M.I. and A.S. analyzed the data. A.S. prepared the original draft and wrote the manuscript with help from M.I.

## Competing interests

The authors declare no competing interests.
