## [Peer Review File · Nature Communications]

FIGNL1 AAA+ ATPase remodels RAD51 and DMC1 filaments in pre-meiotic DNA replication and meiotic recombinationREVIEWER COMMENTS

Reviewer #1 (Remarks to the Author):

The study from Ito et al highlights the role Fignl1 during mouse spermatogenesis and more specifically during meiosis prophase I. It exploits the meiotic recombination to decipher the function of this RAD51 regulator and proposes a role in the disassembly of the recombinase nucleofilament. They nicely combine both detailed analysis of a Fignl1 conditional mutant (and double KO) and biochemistry to sustain their demonstration. Their conclusion is highly similar to that of Baudat's manuscript that also investigated Firm cKO (accompanying this submission) and to the very recently published article from Yu's team (NAR). Nonetheless I consider these independent demonstrations crucial regarding the importance of the topic and the robustness of these studies. Quite interestingly the meiotic mammalian function of FIGNL1 seems to differ noticeably from that earlier described in plants. Below are appended a few comments:

Major comments:

- 1- Meiosis is a process shared by spermatogenesis and oogenesis. Did the authors have any information regarding the role of Fignl1 in the female meiosis? Considering the difference often observed in recombination deficient mutant between sexes, this might allow further understanding of the FIGNL1's role?
- 2- Is FIGNL1 present at recombination nodules? And possibly when?
- 3- Are Fignl1 cKO fertile? Considering these produce sperm it might be. And if so do they transmit the excised allele? I understand the Stra8-cre is an imperfect/incomplete tool. However, gaining confirmation that the cells that behave "normally" are indeed escaping the excision would strengthen the present observations.

Minor comments:

- 1- Introduction line 59, I believe the role of RAD51 in tumorigenesis is a serious matter of controversy and such should be tempered or corrected (see Matos-Rodrigues et al NAR 2021).
- 2- Line 101, it might be misleading to suggest the 'uniqueness' in meiotic recombination relies uniquely on DMC1, especially considering the many species devoid of dmc1. Please rephrase.
- 3- The introduction heavily describes the RAD51 'life' but some elements are not directly useful there and do not help to position the context in a straight manner. A more direct situation of the question could ease the reading.
- 4- Line 161, FIGNL1 being essential for embryonic development is not that easily understood as often it is positive regulator of HR that are proven essential. Should the negative regulation of RAD51 be also considered as a mandatory event to complete HR in somatic cells. More globally, this questions whether the FIGNL1 function described in meiosis is possibly common to mitotic/somatic cells.
- 5- Line 223-225, I do not understand the conclusion. When does the depletion occur precisely is an important matter.
- 6- The ordering of figures is quite unexpected. Fig 2ab are commented first then Figure 3 and then back to Fig2cd. Though I may understand the point wished by authors, a mere mention that Fig2cd will be commented later would help the reader and maintain the -expected- chronological mention of the figures.
- 7- Line 332, is a word missing (not)?
- 8- Line 366, I do not understand what sustains the comment that 'it is unlikely that RAD51 is frequently associated with ongoing DNA replication'.
- 9- The EdU experiment (one hour) may actually reveal some interesting relationship between RAD51 localization and DNA replication. The quantification of such and its comparison to the 10' IdU experiment might be worth either to rule out or suggest a relationship.
- 10- Whether the RAD51 activity at the replication fork is also compromised (or not) in the Fignl1cKO is an important point. A quite simple examination of the duration of the preleptotene stage may already provide a gross answer (alternatively, combining IdU and EdU pulses might here be informative - though I do not expect heavy experiments to be set up to respond to minor comments, considering

the large amount of data already positioned-)

11- Line 426, typo (pun).

12- Line 488-490, Can it be demonstrated that these are indeed cells in which cre-recombination did fail using FIDGNL1 immunostaining?

13- Line 503, explain the use of DeltaN-FIGNL1.

14- An important conclusion of the study is that, in absence of Fignl1, RAD51 and then DMC1 might accumulate on double strand DNA. Considering that such accumulation initiates during the S phase it might be worth tempering such assertion unless the absence of ssDNA might be definitively proven. (IE Fignl1 deficiency might trigger subtle DNA defects even without Spo11 and these may root the RAD51 association to DNA...)

15- The hypothesis that RAD51 accumulation might be an obstacle to invasion (possibly preventing DNA opening) is not considered. Is it due to too few foci?

16- Line 608, do the authors mean 'In human cells/cell lines, ...'

17- Line 853, typo "Brad-ford"

18- Extended data Fig 1, please show the gH2AX channel alone or with a more friendly color. It is quite difficult to recognize the staining below the magenta.

19- Ext data Fig 3c, please zoom in. Change the Y axis. This result is quite intriguing even with a small p value. Whether DMC1 also colocalizes with IdU would be worth investigating.

20- Line 1384 how were chosen the cutoff (10 or 30 foci) ?

Reviewer #2 (Remarks to the Author):

RecA family recombinases function as helical filaments bound to the tracts of ssDNA that form at sites of DNA damage. A vast array of accessory proteins modulate recombinase filaments. Both positive and negative regulation of filament nucleation or stabilization have been described. Regulation of filament assembly/disassembly appears to be particularly elaborate during meiosis with more than a dozen such regulatory proteins previously identified. Here Ito et al. study the role of the AAA-ATPase FIGNL1 in modulating the appearance/disappearance of recombinase complexes during meiotic prophase. FIGNL1 has previously been found to limit RAD51 assembly during mitotic DNA repair in mouse and in meiosis in plants, but has not been studied in meiotic mouse cells previously. The study reveals a role for FIGNL1 in promoting progression of the meiotic recombination pathway. This pro-recombination function appears to be associated with negative regulation of RAD51/DMC1 foci. In addition, a second function of FIGNL1 appears to be preventing the accumulation of complexes of RAD51 and DMC1 caused by direct association with dsDNA. This work will be of broad interest to the meiosis community as well as the homologous recombination community. The paper is clearly written, and the study is well-conceived and executed with a very extensive data set supporting the conclusions of the paper. I think the paper is entirely appropriate for Nature Comm. There are some minor problems with English grammar, and I recommend the journal do a round of copy-editing after re-submission.

I have some concern that the biochemical results might be explained by differences in non-specific protein aggregation of RAD51. The fact that release of free DNA is seen following addition of FIGNL1 to RAD51 DNA mixtures makes this possibility somewhat less likely than if the only impact observed was reduction in RAD51 DNA binding activity, but this issue should be addressed or at least mentioned.

Another issue that I think should be addressed is that mention was made of differences in average RAD51 or DMC1 focus staining intensity. Mutants defective in dissociation of filamentous proteins tend to display foci with increased length or staining intensity. It appears that the differences in staining patterns examined here are restricted to focus number, not focus length or intensity. I think average focus staining intensities should be reported with appropriate interpretation of the results.

The order I present the remaining comments and suggested corrections does not reflect their relative importance.

75. The claim that SWI5-SFR1 suppresses the ATPase activity of RAD51 is false. The papers cited either do not address the issue or show the opposite result. The authors should review their citations and make sure there are no other serious errors.

94. Although mentioned elsewhere in the paper, there is no mention here of the roles of RAD54 family translocases in disassembly of RAD51 and DMC1 from dsDNA recombination intermediates or from off-pathway structures. This would seem to be important given that FIGNL1 appears to have a closely related set of functions.

123. There is no specific reference to results showing that FIGNL1 acts as a positive regulator of RAD51 assembly. If such a claim is correct, the specific result should be mentioned along with appropriate ref.

189. delete "The.." (last word in the line).

190. delete "The.." (last word in the line).

194. Sperm counts were 6-fold lower in the mutant than in the control. Residual sperm might come...

259 the Dmc1

279. This paragraph needs to do a better job of distinguishing prior results with those presented here. For example, this line should be something like "Previous cytological analyses showed...."

285. Here, we analyzed focus formation by MSH4 and found the number of MSH4 foci to be significantly less than that in control cells (50 ± 22 ...).

305. I think it would make more sense for this paragraph to be moved up and placed immediately after the description of the results involving Dmc1 foci.

341. ADD "The", i.e. "The increased number..."

344. "foci that form in the absence of.."

353. "During the meiotic"

379. suggest replacing "on any analyzed stages" with "at each analyzed stage" . Also, the comma should be replaced with a period and the following sentence should start "We observed about 2-fold more..."

412. " ..which are expected to be defective..."

426. "pun" should be "pan"

429 "pseud-sex body" should be "pseudo-sex body"

437. suggest "...chromatin, likely by direct dsDNA binding."

445. replace "sought" with "prompted"

495. This sentence is unclear and should be re-written. What does "its" refer to, RAD51/DMC1 or FIGNL1?

517. Suggest "Fignl1 cKO mice displayed higher levels of RAD51 and DMC1 foci than controls, as well as defective chromosome synapsis."

538. Clarify info on the double mutant mouse.

552. Suggest "RAD51/DMC1 Foci accumulate to higher than normal levels."

560. "during the strand.."

564. suggest replacing entire sentence with "RAD51/DMC1 focus formation normally occurs at sites of ssDNA tracts that are associated with mitotic or meiotic DSBs"

574-575. This sentence is potentially misleading. Previous analysis of RAD54 family mutants suggested that RAD51 and DMC1 loading is not restricted to ssDNA but rather than nascent structures formed on dsDNA are efficiently disassembled under normal conditions such that they are not readily detected. This issue could be side-stepped by saying "On the other hand, detection of RAD51/DMC1 foci normally requires the presence of tracts of ssDNA."

581. References to the Heyer and Chi labs' biochemical work on RAD54 mediated removal of recombinase from dsDNA should be included.

590. "remodelling" should be replaced with "disassembling" or "dismantling".

The method section is unusually long in spite of the fact that essentially all of the methods are very similar to ones published previously. Although it would be convenient to have so much detail here for those wanting to repeat these results, the editors may prefer that the authors shorten this section by citing earlier papers with detailed descriptions of methods along with any needed descriptions of critical revisions of the published protocols.

1334. The "degrees of colocalization" is not what is shown. I think what is plotted is the # of colocalizing foci.

Reviewer #3 (Remarks to the Author):

Ito et al reported the function of FIGNL1 in mouse germ cells. This group previously studied human FIGNL1 in human somatic cells. In this new study, they generated germ cell-specific Fignl1 cKO (using Stra8-Cre) and found that FIGNL1 is essential for meiotic recombination. Specifically, inactivation of FIGNL1 leads to increased number of RAD51 foci and DMC1 foci in prophase I spermatocytes and formation of RAD51 foci and DMC1 foci in preleptotene spermatocytes (pre-meiotic S phase cells), which were absent in wild type. They further showed that RAD51 foci and DMC1 foci were still formed

in Figl1 cKO spermatocytes in the absence of SPO11. They also showed that inactivation of FIGNL1 rescues the RAD51 formation defects in Swsap1 mouse mutant. Biochemical experiments showed that human FIGNL1 Δ disrupts RAD51-dsDNA complex.

This study demonstrates an essential role of FIGNL1 in male meiosis. It provides mechanistic insights. The study is comprehensive. The conclusions are supported by the data. The manuscript is well written. It will be of interest to readers in the field of meiosis and recombination.

Major concerns:

1) Title needs to be revised. The DNA replication prior to meiosis should be called pre-meiotic DNA replication not meiotic. The germ cell could undergo either mitosis or meiosis. Suggestion: FIGNL1 AAA+ ATPase remodels RAD51 and DMC1 filaments in pre-meiotic DNA replication and meiotic recombination. "the meiotic S-phase" in the abstract needs to be changed to "the pre-meiotic S-phase".

2) Figure 2 and Figure 3 need to be switched. As is, Figure 3 is about RPA and MSH4. Figures 2 and 4 are about RAD51 and DMC1. Therefore, Figure 3 disrupts the logic flow for the manuscript. Accordingly, the result section "Compromised meiotic recombination in Figl1 cKO" needs to be moved up, before the section titled "FIGNL1 promotes the disassembly of DMC1 in male meiosis".

3) RAD51 ChIP (SSDS) was performed in Figl1 cKO testis but not in Spo11 Figl1 double knockout testis. It is expected that RAD51 binds to meiotic DSB hotspots in Figl1 cKO testis, because SPO11 is intact. In addition, the SSDS protocol only identifies ssDNA. The RAD51 SSDS in Figl1 cKO testis was not really informative. Such SSDS analysis in Spo11 Figl1 double knockout testis would be more informative. It will address whether RAD51 foci still correspond to ssDNA (probably not). This point needs to be considered or at least discussed.

4) Immunofluorescence of FIGNL1 needs to be performed in testis sections. This will show the expression of FIGNL1 among different germ cells and testicular somatic cells and its subcellular localization. Such information can be provided as supplementary.

Minor point:

The role of FIGNL1 in female meiosis is unknown, because the Stra8-Cre transgene used in this study is not active in female germ cells. This can be mentioned for the benefit of readers.

Line184: "in spermatogonia (mitotic stem cells)". Please delete mitotic stem cells. Stra8-Cre is expressed in differentiating spermatogonia, which are not mitotic stem cells.

Line 426: please change "pun-nuclear" to "pan-nuclear".

Responses to reviewers:

We do appreciate for constructive comments on our paper by the reviewers, which are helpful for the improvement of our paper. We revised the paper based on the advice from the editor and reviewers. Our responses below are shown in blue.

Reviewer #1 (Remarks to the Author):

The study from Ito et al highlights the role *Figl1* during mouse spermatogenesis and more specifically during meiosis prophase I. It exploits the meiotic recombination to decipher the function of this RAD51 regulator and proposes a role in the disassembly of the recombinase nucleofilament. They nicely combine both detailed analysis of a *Figl1* conditional mutant (and double KO) and biochemistry to sustain their demonstration. Their conclusion is highly similar to that of Baudat's manuscript that also investigated *Firm* cKO (accompanying this submission) and to the very recently published article from Yu's team (NAR). Nonetheless I consider these independent demonstrations crucial regarding the importance of the topic and the robustness of these studies. Quite interestingly the meiotic mammalian function of FIGNL1 seems to differ noticeably from that earlier described in plants. Below are appended a few comments:

Major: comments:

1- Meiosis is a process share by spermatogenesis and oogenesis. Did the authors have any information regarding the role of *Figl1* in the female meiosis? Considering the difference often observed in recombination deficient mutant between sexes, this might allow further understanding the FIGNL1's role?

-The role of FIGNL1 in female meiosis is what we want to know now, but unfortunately, the efficiency of *Stra8-Cre*-mediated cKO of *Figl1* in oocytes is quite lower compared to that in spermatocytes described in our paper (~80%). We are still working to figure out the best *Cre* transgenes to characterize the role of FIGNL1 in female meiosis. On the other hand, a recent NAR paper by Zhang et al showed that conditional KO of *FLIP/FIRRM*, a partner of FIGNL1, leads to infertility in both male and female mice, suggesting that the FIGNL1-FIRRM complex is essential for both spermatogenesis and oogenesis. We mentioned this point in the Discussion section by citing this paper (pages 17-18, lines 584-590).

2- Is FIGNL1 present at recombination nodules? And possibly when?

-FIGNL1 immunostaining in testicular cell squashes showed pan-nucleus staining in SYCP3-positive spermatocytes, rather than specific enrichment on chromosome axes where recombination nodules localize (Supplementary Figure 1b). Immunolocalization of FIGNL1 on chromosome spreads would answer this question the best, but unfortunately, FIGNL1 immunostaining with our homemade and commercially available antibodies didn't work on chromosome spreads in our hands (unpublished results). From our current observation in testicular cell squashes, we suspect that FIGNL1 exists in the entire nucleus throughout meiotic prophase I with a dynamic nature, though this doesn't rule out a possibility that FIGNL1 transiently localizes to recombination nodules to function in RAD51/DMC1 disassembly.

3- Are *Figl1* cKO fertile? Considering these produce sperm it might be. And if so do they transmit the excised allele? I understand the *Stra8-cre* is an imperfect/incomplete tool. However, gaining confirmation that the cells that behave “normally” are indeed escaping the excision would strengthen the present observations.

-This is an important point. *Figl1* cKO is expected to be fertile, but we haven't tested it or analyzed the transmission of the excised allele because 1) a substantial number of cells in *Figl1* cKO escaped *Stra8-Cre* excision and produced sperms, and 2) the excised *Figl1^Δ* allele can be transmitted even from *Figl1^{flox/Δ} Stra8-Cre⁺* spermatocytes that failed the excision. The frequency of the transmission of the excised allele might be informative, but not definitive. We showed that nearly all (99%) pachytene/diplotene cells with γ H2AX staining restricted to the sex body in *Figl1* cKO testes are FIGNL1-positive cells that escaped the excision (Figure 1h; Supplementary Figure 1b). We believe that this data would be sufficient to conclude that *Figl1* KO cells are all arrested before pachynema and only cells that escaped the excision progress further and behave normally to complete spermatogenesis. For clarification, we added this description in the Result section (page 7, lines 206-208).

Minor comments:

1- Introduction line 59, I believe the role of RAD51 in tumorigenesis is a serious matter of controversy and such should be tempered or corrected (see Matos-Rodrigues et al NAR 2021).

-We deleted this sentence according to point #3.

2- Line 101, it might be misleading to suggest the ‘uniqueness’ in meiotic recombination relies uniquely on DMC1, especially considering the many species devoid of *dmc1*. Please rephrase.

-Agreed and rephrased as “DMC1, is critical for homology search in meiotic recombination in many species” (page 4, line 88).

3- The introduction heavily describes the RAD51 ‘life’ but some elements are not directly useful there and do help to position the context in a straight manner. A more direct situation of the question could ease the reading.

-We shortened the Introduction in a way to clarify our questions in the paper.

4- Line 161, FIGNL1 being essential for embryonic development is not that easily understood as often it is positive regulator of HR that are proven essential. Should the negative regulation of RAD51 be also considered as a mandatory event to complete HR in somatic cells. More globally, this questions whether the FIGNL1 function described in meiosis is possibly common to mitotic/somatic cells.

-We believe that FIGNL is a “positive” regulator in HR with a negative role in RAD51 assembly in a context-dependent manner. Previous and recent reports clearly showed that FIGNL1 and its partner FIRMM promote HR and ICL repair (Yuan and Chen, *PNAS* 2013; Pinedo-Carpio et al., *Science Advance* 2023; Zhou et al., *Cell Reports* 2023), suggesting a positive role of the FIGNL1-FIRRM complex in HR-mediated repair. Given this positive role of FIGNL1 in the HR pathway, it is not surprising that *Figl1* KO shows

embryonic lethality as KOs of other HR genes. We stressed the positive role of FIGNL1 in Abstract and Introduction sections (page 2, lines 39-40; page 4, lines 108-109).

5- Line 223-225, I do not understand the conclusion. When does the depletion occurs precisely is an important matter.

-We rewrote this sentence as "*Figl1*^{Δ/Δ} cells are fully lost before pachynema", given that γH2AX staining pattern on testicular cell squashes allows us to distinguish cells before pachynema (leptonema/zygonema) from those after pachynema (pachynema/diplonema) (page 7, line 206).

6- The ordering of figures is quite unexpected. Fig 2ab are commented first then Figure 3 and then back to Fig2cd. Though I may understand the point wished by authors, a mere mention that Fig2cd will be commented later would help the reader and maintain the -expected- chronological mention of the figures.

-We changed the order of Figures 2 and 3 according to reviewer #3's suggestion. As a result, Figure 2cd is commented immediately after Figure 2ab.

7- Line 332, is a word missing (not)?

-Rephrased as "yet to be elongated".

8- Line 366, I do not understand what sustains the comment that 'it is unlikely that RAD51 is frequently associated with ongoing DNA replication'.

-We rewrote this as "it is unlikely that RAD51 is associated with ongoing DNA replication forks" (we added "forks" and removed "frequently" from the original sentence), as the degree of RAD51 colocalization with IdU (10' labeling), which is expected to correspond to ongoing DNA replication forks, is pretty low (Supplementary Figure 4c).

9- The EdU experiment (one hour) may actually reveal some interesting relationship between RAD51 localization and DNA replication. The quantification of such and its comparison to the 10' IdU experiment might be worth either to rule out or suggest a relationship.

-We quantified the degree of colocalization between RAD51 and EdU (60' labeling) and compared it with the 10' IdU labeling in Supplementary Figure 4d. As expected, RAD51 colocalized more frequently with the 60' EdU than the 10' IdU. Considering this data and point #8, we rewrote the last sentence of this paragraph as "it is unlikely that RAD51 is associated with ongoing DNA replication forks" (page 11, lines 357-358).

10- Whether the RAD51 activity at the replication fork is also compromised (or not) in the *Figl1*cKO is an important point. A quite simple examination of the duration of the preleptonema stage may already provide a gross answer (alternatively, combining IdU and EdU pulses might here be informative -though I do not expect heavy experiments to be sat up to respond to minor comments, considering the large amount of data already positioned-)

-To address the question, we simply counted the number of RPA2 foci, which are expected to be associated with ssDNA on replication forks, and IdU foci in preleptonema (pre-meiotic S-phase) and found that these numbers were comparable between *Figl1*

cKO and controls (Supplementary Figure 6). Combined with the result that cells progress into meiotic prophase I in *Figl1* cKO, we believe that pre-meiotic DNA replication in *Figl1* cKO is largely normal. Of course, the DNA fiber assay is required to detect subtle changes in replication fork speeds or origin firing. However, in our experimental system, the assessment of pre-meiotic DNA replication by the DNA fiber assay in cKO testes would be obscured by a substantial number of DNA molecules from cells without the excision (i.e. wild-type). A recent NAR paper by Zhang et al showed, by the DNA fiber assay, that *FIGNL1* KO (not cKO) human cells are defective in replication restart. This suggests that *Figl1* deficiency also affects the RAD51 activity at the replication fork under replication stress. This point is described in the Result section (pages 11-12, lines 371-381).

11- Line 426, typo (pun).
-Fixed.

12- Line 488-490, Can it be demonstrated that these are indeed cells in which cre-recombination did fail using FIDGNL1 immunostaining?
-As mentioned in major point #2, FIGNL1 immunostaining didn't work on chromosome spreads in our hands. It worked on testicular cell squashes, but it's hard to distinguish mid zygotene cells from late zygotene cells on testicular cell squashes.

13- Line 503, explain the use of DeltaN-FIGNL1.
-Since the full-length human FIGNL1 protein was insoluble when expressed in *E. coli*, we used FINGL Δ N, which was soluble. This is mentioned on page 16, lines 519-520.

14- An important conclusion of the study is that, in absence of *Figl1*, RAD51 and then DMC1 might accumulate on double strand DNA. Considering that such accumulation initiates during the S phase it might be worth tempering such assertion unless the absence of ssDNA might be definitively proven. (IE *Figl1* deficiency might trigger subtle DNA defects even without *Spo11* and these may root the RAD51 association to DNA...)
-Thank you for pointing this out. The number of RPA2 foci, which are expected to correspond to ssDNAs, in early preleptonema (early pre-meiotic S-phase) were comparable between *Figl1* cKO and controls, as mentioned above (point #10). To exclude a possibility that a subtle change in ssDNA levels caused by *Figl1* deficiency was masked by a large number of replication fork-associated ssDNAs, we also quantified the number of RPA2 foci in early leptonema, soon after pre-meiotic DNA replication-dependent RPA2 foci disappeared, in *Spo11* KO and *Spo11* KO *Figl1* cKO (i.e. without SPO11-mediated meiotic DSBs) (Supplementary Figure 8f). The number of RPA2 foci was comparable between *Spo11* KO and *Spo11* KO *Figl1* cKO, suggesting that *Figl1* deficiency doesn't induce DNA lesions that are marked by RPA, at least on the cytological level. To support our conclusion that RAD51 (and DMC1) accumulate on dsDNA in *Figl1* cKO, we also ran RAD51 ChIP-SSDS in *Spo11* KO and *Spo11* KO *Figl1* cKO testes (Supplementary Figure 10). Most of RAD51 peaks in *Spo11* KO *Figl1* cKO testes were commonly seen in *Spo11* KO testes, suggesting that *Figl1* deficiency doesn't induce RAD51-ssDNAs to specific genomic regions in a population analysis. Although these data won't exclude possibilities that cytologically undetectable levels of subtle DNA lesions are

triggered by *Figl1* deficiency and/or distribution of those lesions on the genome were variable (random) among cells (thus undetectable in the population analysis) with preferential binding of RAD51 to the ssDNAs, we believe that our data support the idea that most of RAD51 (and DMC1) foci in *Figl1* cKO are on intact dsDNAs.

15- The hypothesis that RAD51 accumulation might be an obstacle to invasion (possibly preventing DNA opening) is not considered. Is it due to too few foci?

-We mentioned this possibility in the Discussion section as “It is likely that FIGNL1 disrupts not only RAD51 but also DMC1 filaments on recombination intermediates for subsequent DNA synthesis, though other possibilities including that proper disassembly of RAD51 and DMC1 filaments promotes strand invasion won’t be excluded” (page 17, lines 565-567).

16- Line 608, do the authors mean ‘In human cells/cell lines, ...’

-Yes, and fixed.

17- Line 853, typo “Brad-ford”

-Fixed.

18- Extended data Fig 1, please show the γ H2AX channel alone or with a more friendly color. It is quite difficult to recognize the staining below the magenta.

-We modified the Supplementary Figure 1 (Extended Data Figure 1) to show the γ H2AX channel alone and changed the color of γ H2AX in merged images from blue to green by removing the FIGNL1 channel (showing the FIGNL1 channel alone should be fine to show its positive staining in spermatocytes).

19- Ext data Fig 3c, please zoom in. Change the Y axis. This result is quite intriguing even with a small p value. Whether DMC1 also colocalizes with IdU would be worth investigating.

-We changed the Y axis so that each plot looks clear. We also analyzed the colocalization of DMC1 and EdU (10’ labeling) in Supplementary Figure 5 and described this in the Result section (page 11, lines 363-366).

20- Line 1384 how were chosen the cutoff (10 or 30 foci) ?

-The cutoff was chosen so that nearly all cells in the controls were below the cutoff. DMC1 foci in preleptotene (pre-meiotic S) *Figl1* cKO spermatocytes are a bit dim and the threshold of DMC1-focus intensity to include them resulted in a higher number of DMC1 foci than RAD51 foci in controls, thus higher cutoff for DMC1 than RAD51.

Reviewer #2 (Remarks to the Author):

RecA family recombinases function as helical filaments bound to the tracts of ssDNA that form at sites of DNA damage. A vast array of accessory proteins modulate recombinase filaments. Both positive and negative regulation of filament nucleation or stabilization have been described. Regulation of filament assembly/disassembly appears to be particularly elaborate during meiosis with more than a dozen such regulatory proteins

previously identified. Here Ito et al. study the role of the AAA-ATPase FIGNL1 in modulating the appearance/disappearance of recombinase complexes during meiotic prophase. FIGNL1 has previously been found to limit RAD51 assembly during mitotic DNA repair in mouse and in meiosis in plants, but has not been studied in meiotic mouse cells previously. The study reveals a role for FIGNL1 in promoting progression of the meiotic recombination pathway. This pro-recombination function appears to be associated with negative regulation of RAD51/DMC1 foci. In addition, a second function of FIGNL1 appears to be preventing the accumulation of complexes of RAD51 and DMC1 caused by direct association with dsDNA. This work will be of broad interest to the meiosis community as well as the homologous recombination community. The paper is clearly written, and the study is well-conceived and executed with a very extensive data set supporting the conclusions of the paper. I think the paper is entirely appropriate for Nature Comm. There are some minor problems with English grammar, and I recommend the journal do a round of copy-editing after re-submission.

I have some concern that the biochemical results might be explained by differences in non-specific protein aggregation of RAD51. The fact that release of free DNA is seen following addition of FIGNL1 to RAD51 DNA mixtures makes this possibility somewhat less likely than if the only impact observed was reduction in RAD51 DNA binding activity, but this issue should be addressed or at least mentioned.

-We mentioned this possibility as “Alternatively, RAD51-ssDNA *in vitro* might form an aggregate more than RAD51-dsDNA, which may affect the accessibility by FIGNL1 protein to disrupt the complexes” (Page 16, line 530-532).

Another issue that I think should be addressed is that mention was made of differences in average RAD51 or DMC1 focus staining intensity. Mutants defective in dissociation of filamentous proteins tend to display foci with increased length or staining intensity. It appears that the differences in staining patterns examined here are restricted to focus number, not focus length or intensity. I think average focus staining intensities should be reported with appropriate interpretation of the results.

-This is an important point, and we quantified the intensities of RAD51 and DMC1 foci. Actually, the intensities of RAD51 and DMC1 foci in *Figl1* cKO were lower than those in controls (Supplementary Figure 3). We interpret that, in *Figl1* cKO spermatocytes, more sites for the accumulation of RAD51/DMC1 molecules are present, fewer molecules of RAD51/DMC1 protein are on each accumulation site because of a limited number of RAD51/DMC1 protein in a cell. We described this point in the Result section (page 10, lines 305-310).

The order I present the remaining comments and suggested corrections does not reflect their relative importance.

75. The claim that SWI5-SFR1 suppresses the ATPase activity of RAD51 is false. The papers cited either do not address the issue or show the opposite result. The authors should review their citations and make sure there are no other serious errors.

-Since reviewer #1 asked to shorten the Introduction, we deleted sentences describing SWI5-SFR1.

94. Although mentioned elsewhere in the paper, there is no mention here of the roles of RAD54 family translocases in disassembly of RAD51 and DMC1 from dsDNA recombination intermediates or from off-pathway structures. This would seem to be important given that FIGNL1 appears to have a closely related set of functions.

-We added a brief description of RAD54 here as “RAD54 translocase also disassembles RAD51 from dsDNA in recombination intermediates” (page 3, lines 79-80).

123. There is no specific reference to results showing that FIGNL1 acts as a positive regulator of RAD51 assembly. If such a claim is correct, the specific result should be mentioned along with appropriate ref.

-We corrected this sentence as “Despite its positive role in homologous recombination, FIGNL1 functions as a negative regulator for RAD51 assembly” (page 4, lines 108-109).

189. delete “The..” (last word in the line).

-Done.

190. delete “The..” (last word in the line).

-Done.

194. Sperm counts were 6-fold lower in the mutant than in the control. Residual sperm might come...

-Fixed.

259 the Dmc1

-Since we worked on mouse DMC1 protein, we prefer using “DMC1” rather than “Dmc1”.

279. This paragraph needs to do a better job of distinguishing prior results with those presented here. For example, this line should be something like “Previous cytological analyses showed....”

-Fixed as suggested.

285. Here, we analyzed focus formation by MSH4 and found the number of MSH4 foci to be significantly less than that in control cells (50 ± 22 ...).

-Fixed by incorporating the suggestion as “We analyzed MSH4 focus formation and found that the number of MSH4 foci in *Figl1* cKO was lower than in controls” (page 8, lines 246-247).

305. I think it would make more sense for this paragraph to be moved up and placed immediately after the description of the results involving Dmc1 foci.

-We changed the order of Figures 2 and 3 according to reviewer #3’s suggestion. As a result, the description of RAD51 was placed immediately after that of DMC1.

341. ADD “The”, i.e. “The increased number...”

-Done.

344. “foci that form in the absence of..”

-Fixed.

353. “During the meiotic”

-Fixed.

379. Suggest replacing “on any analyzed stages” with “at each analyzed stage” . Also, the comma should be replaced with a period and the following sentence should start “We observed about 2-fold more...”

-We replaced “on any analyzed stages” with “at each analyzed stage”, and rewrote the following sentence as “we observed ~2-fold more...” as suggested, but kept the comma as it is because we didn’t think this replacement is necessary (page 12, lines 384-386).

412. “ ..which are expected to be defective...”

-Fixed.

426. “pun” should be “pan”

-Fixed.

429 “pseud-sex body” should be “pseudo-sex body”

-Fixed.

437. Suggest “...chromatin, likely by direct dsDNA binding.”

-Thanks for the suggestion. Here we meant that RAD51-bound meiotic chromatin that DMC1 preferentially binds to is likely dsDNA, rather than saying that DMC1 directly binds to dsDNAs. To clarify that the data in the absence of SPO11/meiotic DSBs described here support the observation in *Figl1* preleptonema (pre-meiotic S) described in a paragraph immediately before, we rewrote this sentence as “This supports the idea that DMC1 preferentially binds to RAD51-bound meiotic chromatin, which is most likely dsDNAs in the absence of meiotic DSB formation” (pages 13-14, lines 443-445).

445. Replace “sought” with “prompted”

-Done.

495. This sentence is unclear and should be re-written. What does “its” refer to, RAD51/DMC1 or FIGNL1?

-We rewrote this sentence as “It is likely that the activity of FIGNL1 to suppress RAD51/DMC1 assembly on intact dsDNAs is independent of SWSAP1-mediated stable assembly of RAD51/DMC1 on DSB-induced ssDNAs (page 15, lines 509-511).

517. Suggest “Figl1 cKO mice displayed higher levels of RAD51 and DMC1 foci than controls, as well as defective chromosome synapsis. “

-Fixed.

538. Clarify info on the double mutant mouse.

-We rewrote this sentence to clarify the information on the *Rad54 Rad54b* double KO mice (page 17, lines 557-560).

552. Suggest “RAD51/DMC1 Foci accumulate to higher than normal levels.”

-Fixed as “RAD51/DMC1 foci accumulate in a higher number than normal” (page 17, line 575).

560. “during the strand..”

-We are sorry that we can not understand what this means since we used the same words in the main text.

564. Suggest replacing entire sentence with “RAD51/DMC1 focus formation normally occurs at sites of ssDNA tracts that are associated with mitotic or meiotic DSBs”

-Fixed and the following sentence was also modified (page 18, lines 594-595).

574-575. This sentence is potentially misleading. Previous analysis of RAD54 family mutants suggested that RAD51 and DMC1 loading is not restricted to ssDNA but rather than nascent structures formed on dsDNA are efficiently disassembled under normal conditions such that they are not readily detected. This issue could be side-stepped by saying “On the other hand, detection of RAD51/DMC1 foci normally requires the presence of tracts of ssDNA.”

-Agreed and fixed.

581. References to the Heyer and Chi labs’ biochemical work on RAD54 mediated removal of recombinase from dsDNA should be included.

-We corrected the corresponding paragraph by citing additional biochemical papers (page 18, lines 618-620).

590. “remodelling” should be replaced with “disassembling” or “dismantling”.

-Replaced with disassembling.

The method section is unusually long in spite of the fact that essentially all of the methods are very similar to ones published previously. Although it would be convenient to have so much detail here for those wanting to repeat these results, the editors may prefer that the authors shorten this section by citing earlier papers with detailed descriptions of methods along with any needed descriptions of critical revisions of the published protocols.

-We shortened “Spermatocytes-chromosome spreads” and “RAD51 ChIP-SSDS” in the method section by deleting too much details (pages 20-21, lines 685-726; page 24-25, lines 805-856).

1334. The “degrees of colocalization” is not what is shown. I think what is plotted is the # of colocalizing foci.

-The number of colocalizing foci is plotted in Figure 3e and the degrees of colocalization are plotted in Figure 3f. A sentence you pointed corresponds to Figure 3f so it would be fine.

Reviewer #3 (Remarks to the Author):

Ito et al reported the function of FIGNL1 in mouse germ cells. This group previously studied human FIGNL1 in human somatic cells. In this new study, they generated germ cell-specific *Figl1* cKO (using *Stra8-Cre*) and found that FIGNL1 is essential for meiotic recombination. Specifically, inactivation of FIGNL1 leads to increased number of RAD51 foci and DMC1 foci in prophase I spermatocytes and formation of RAD51 foci and DMC1 foci in preleptotene spermatocytes (pre-meiotic S phase cells), which were absent in wild type. They further showed that RAD51 foci and DMC1 foci were still formed in *Figl1* cKO spermatocytes in the absence of SPO11. They also showed that inactivation of FIGNL1 rescues the RAD51 formation defects in *Swsap1* mouse mutant. Biochemical experiments showed that human FIGNL1 Δ N disrupts RAD51-dsDNA complex.

This study demonstrates an essential role of FIGNL1 in male meiosis. It provides mechanistic insights. The study is comprehensive. The conclusions are supported by the data. The manuscript is well written. It will be of interest to readers in the field of meiosis and recombination.

Major concerns:

1) Title needs to be revised. The DNA replication prior to meiosis should be called pre-meiotic DNA replication not meiotic. The germ cell could undergo either mitosis or meiosis. Suggestion: FIGNL1 AAA+ ATPase remodels RAD51 and DMC1 filaments in pre-meiotic DNA replication and meiotic recombination. “the meiotic S-phase” in the abstract needs to be changed to “the pre-meiotic S-phase”.

-We changed the title as suggested and rephrased all “meiotic S-phase” in the manuscript to “pre-meiotic S-phase”.

2) Figure 2 and Figure 3 need to be switched. As is, Figure 3 is about RPA and MSH4. Figures 2 and 4 are about RAD51 and DMC1. Therefore, Figure 3 disrupts the logic flow for the manuscript. Accordingly, the result section “Compromised meiotic recombination in *Figl1* cKO” needs to be moved up, before the section titled “FIGNL1 promotes the disassembly of DMC1 in male meiosis”.

-Thank you for the suggestion. We changed the order of Figures as suggested, with some minor changes in the description of the initial part of each section.

3) RAD51 ChIP (SSDS) was performed in *Figl1* cKO testis but not in *Spo11 Figl1* double knockout testis. It is expected that RAD51 binds to meiotic DSB hotspots in *Figl1* cKO testis, because SPO11 is intact. In addition, the SSDS protocol only identifies ssDNA. The RAD51 SSDS in *Figl1* cKO testis was not really informative. Such SSDS analysis in *Spo11 Figl1* double knockout testis would be more informative. It will address whether RAD51 foci still correspond to ssDNA (probably not). This point needs to be considered or at least discussed.

-Thanks for the suggestion. We ran the RAD51 SSDS in *Spo11 KO Figl1* cKO (i.e. *Spo11 Figl1* double knockout) testes as well as *Spo11* KO testes (two replicates each). As expected, no RAD51 enrichment was detected at meiotic DSB hotspots in *Spo11 KO Figl1* cKO and *Spo11* KO testes. Most of the chromosomal regions with high RAD51

SSDS signals in *Spo11* KO *Figl1* cKO testes were common to *Spo11* KO testes (Supplementary Figure 10). This indicates that the levels of RAD51-ssDNA are comparable between *Spo11* KO *Figl1* cKO and *Spo11* KO testes, at least in a population analysis, and also suggests that RAD51 foci in *Spo11* KO *Figl1* cKO spermatocytes correspond to dsDNAs. We mentioned this point in the Result section (page 14, lines 461-469).

4) Immunofluorescence of FIGNL1 needs to be performed in testis sections. This will show the expression of FIGNL1 among different germ cells and testicular somatic cells and its subcellular localization. Such information can be provided as supplementary.

-Although FIGNL1 immunostaining didn't work in testis sections, it worked in testicular cell squashes in our hands, so we added cell squash images to show the expression of FIGNL1 in SYCP3-positive spermatocytes and PLZF-positive spermatogonia, but not in Sertoli cells or round spermatids, which are distinguished by DAPI staining, in Supplementary Figure 1a. We also mentioned this point in the Result section (page 6, lines 179-181).

Minor point:

The role of FIGNL1 in female meiosis is unknown, because the *Stra8*-Cre transgene used in this study is not active in female germ cells. This can be mentioned for the benefit of readers.

-We mentioned this point in the Discussion section and cited a recent NAR paper by Yu and collages that suggests the common role of the FIGNL1-FIRRM complex in male and female meiosis (pages 17-18, lines 584-590).

Line184: "in spermatogonia (mitotic stem cells)". Please delete mitotic stem cells. *Stra8*-Cre is expressed in differentiating spermatogonia, which are not mitotic stem cells.

-Fixed.

Line 426: please change "pun-nuclear" to "pan-nuclear".

-Fixed.

REVIEWERS' COMMENTS

Reviewer #1 (Remarks to the Author):

Authors did nicely improve the manuscript and addressed all issues raised. Their work broadens our understanding of FIGNL1 and meiotic recombination.

Reviewer #2 (Remarks to the Author):

The revisions are satisfactory.

Reviewer #3 (Remarks to the Author):

My concerns have been addressed. In particular, the authors added RAD51 SSDS data from Figl1 cKO Spo11 double mutant and Spo11 KO testes.

Point-by-point responses to reviewers:

We do appreciate for constructive comments on our paper by the reviewers again, which are helpful for the improvement of our paper. For the new Supplementary Fig. 6, a shows images from 60' EdU labeling experiments and c shows quantification from 10' IdU labeling experiments. For clarification, we added a description as "Representative images of spermatocytes-chromosome spreads are shown in Supplementary Fig. 4a" in the legends for Supplementary Fig. 6c and also described labeling time for all of EdU and IdU labeling experiments in the corresponding figure legends.

Reviewer #1 (Remarks to the Author):

Authors did nicely improve the manuscript and addressed all issues raised. Their work broadens our understanding of FIGNL1 and meiotic recombination.

Reviewer #2 (Remarks to the Author):

The revisions are satisfactory.

Reviewer #3 (Remarks to the Author):

My concerns have been addressed. In particular, the authors added RAD51 SSSS data from Fignl1 cKO Spo11 double mutant and Spo11 KO testes.